# Potential plant extinctions with the loss of the Pleistocene mammoth steppe

Jérémy Courtin[1], Kathleen R. Stoof-Leichsenring[1], Simeon Lisovski[1], Ying Liu[1], Inger Greve Alsos[2], Boris K. Biskaborn[1], Bernhard Diekmann[1], Martin Melles[3], Bernd Wagner[3], Luidmila Pestryakova[4], James Russell[5], Yongsong Huang[5] & Ulrike Herzschuh[1,6,7] ✉

During the Pleistocene-Holocene transition, the dominant mammoth steppe ecosystem across northern Eurasia vanished, in parallel with megafauna extinctions. However, plant extinction patterns are rarely detected due to lack of identifiable fossil records. Here, we introduce a method for detection of plant taxa loss at regional (extirpation) to potentially global scale (extinction) and their causes, as determined from ancient plant DNA metabarcoding in sediment cores (*sed*aDNA) from lakes in Siberia and Alaska over the past 28,000 years. Overall, potential plant extinctions track changes in temperature, in vegetation, and in megafauna extinctions at the Pleistocene-Holocene transition. Estimated potential plant extinction rates were 1.7–5.9 extinctions per million species years (E/MSY), above background extinction rates but below modern estimates. Major potential plant extinction events were detected around 17,000 and 9000 years ago which lag maximum vegetation turnover. Our results indicate that herbaceous taxa and taxa contributing less to beta diversity are more vulnerable to extinction. While the robustness of the estimates will increase as DNA reference libraries and ancient *sed*aDNA data expand, the available data support that plants are more resilient to environmental changes than mammals.

Human-induced climate change and habitat fragmentation are causing widespread damage to ecosystems, triggering species extinctions at a global scale[1–4]. Estimated extinction rates over the past 500 years range between 150 and 260 per million species years (E/MSY), significantly exceeding the 0.1–1 E/MSY background extinction rates[3]. The impact of species extinctions extends across all taxonomic groups of Eukaryotes, with varying degrees of vulnerability observed[5]. Since 1900, mammals have faced the greatest extinction pressure, with 243 E/MSY[3], of course, influenced by taxonomic biases arising from different research efforts across species groups[3,6–9]. Emerging evidence

highlights the underappreciated susceptibility of certain taxonomic groups to increased extinction risks[3,10,11]. Among the less investigated taxa, plant species are more prevalent[12–14]. Notably, 20% of plant species are currently classified as threatened (IUCN), a proportion similar to that of threatened mammal species[15,16]. Extinction rates of plants range from 18 to 26 E/MSY, up to 500 times the background rate[17]. However, uncertainties are particularly high for plants because, among others, their potential dormancy in the seed bank.

Extinctions are more common in regions with high species diversity[5,18]. They occur globally, but only 1–2 plant extinctions are

[1]Polar Terrestrial Environmental Systems, Alfred Wegener Institute Helmholtz Centre for Polar and Marine Research, Potsdam, Germany. [2]The Arctic University Museum of Norway, UiT - The Arctic University of Norway, Tromsø, Norway. [3]Institute of Geology and Mineralogy, University of Cologne, Cologne, Germany. [4]Institute of Natural Sciences, North-Eastern Federal University of Yakutsk, Yakutsk, Russia. [5]Department of Earth, Environmental and Planetary Sciences, Brown University, Providence, RI, USA. [6]Institute of Environmental Science and Geography, University of Potsdam, Potsdam, Germany. [7]Institute of Biology and Biochemistry, University of Potsdam, Potsdam, Germany. ✉e-mail: ulrike.herzschuh@awi.de

known from boreal-arctic regions[17]. Low arctic diversity may to some extent originate from recurrent extinction waves related to severe glacial-interglacial climate and environmental change, providing a suitable testbed to examine species extinctions in the context of natural climate variability. This applies in particular to the transition from the last glacial to the Holocene between 19 and 9 ka BP for which intense investigations revealed severe extinctions among megafauna[19–23]. In parallel to the megafauna loss, the well-established mammoth steppe ecosystem disappeared with no modern analogue but some relics in the Central Asian mountain ranges[24–31]. This ecosystem was characterised by a rich mosaic of woodlands and grassland covering much of the unglaciated northern hemisphere that has persisted through several glacial and interglacial stages[31,32]. Despite the supposed role of Pleistocene megafauna as keystone species, interacting extensively and shaping their environment, only one plant taxon, a *Picea* (spruce) species present in North America, has been reported extinct in the area during the late Pleistocene[33] while further extinctions were reported for tree species in northern Europe during the Quaternary[34]. The surprising lack of any plant-mammalia co-extinction may point to our misconception of the role of large mammal herbivores in maintaining grassy ecosystems as proposed for eastern Beringia[35] and Tibet[36] or derives from our poor understanding of the eco-evolutionary traits that support plant taxa survival, or it simply originates from our methodological limitations to record extinct plant taxa.

Connecting specific traits to extinction remains elusive, emphasising the necessity for recent investigations to clarify these relationships. Recent observations indicate that plant taxa inhabiting the periphery of their native ranges, regionally rare species and habitat specialists face heightened extinction risks, while broader-ranging clades exhibit greater resilience to climatic shifts[37]. However, the realisation that rarity does not consistently heighten extinction risk, and that introductions and speciation may intersect with extinction, further complicates our understanding[38,39]. Additionally, factors like plant size and seed production dynamics play crucial roles in determining extinction probabilities[40]. Despite these insights, discerning a clear phylogenetic pattern in plant extinctions remains challenging[40,41]. This may be attributed to the 'shotgun extinction' principle, where extinctions occur predominantly at the species level across various phyla, necessitating species-level detection to infer extinction events accurately[42]. Plant macrofossils are scarce[43], and microfossils rarely possess sufficient diagnostic characters to allow for species description[44], unlike mammals, where even full carcasses have been found allowing detailed descriptions of extinct species (i.e. ref. 45). Plant taxonomy heavily relies on floral characteristics, which are rarely well preserved, traditionally limiting the study of past plant extinctions making direct comparisons of plant extinction at genus of family level especially during mass extinction events, distinct from the Pleistocene/Holocene transition[46].

The investigation of plant extinction (the global disappearance) or extirpation (spatially restricted disappearance) requires a methodologically consistent assessment of taxa occurrences across space and time, ideally at the species level[3,47]. Currently, there are no multisite time-series datasets available which cover the loss of the mammoth steppe at a subcontinental scale with the required taxonomic precision. Synthesised spatiotemporal pollen datasets, for instance, typically only identify plant taxa to the family or genus level (e.g. ref. 44). Plant macrofossils, although often allowing identification to species level, are rare due to degradation in the absence of optimal preservation conditions[48]. Metabarcoding, utilising the chloroplast *trn*L (UAA) intron as a marker to analyse sedimentary ancient DNA (*sed*aDNA), can provide the required taxonomic resolution for assessing changes in species richness[27,49,50] and the plant species pool[51]. However, like all genetic methods, metabarcoding relies on taxonomic assignments against DNA reference databases, which often have limited representation of rare taxa, and with few exceptions[52], do not

contain sequences of extinct taxa[53,54]. To improve the detection of extinct and rare taxa absent from databases, new concepts are needed, as current plant metabarcoding analyses mostly only consider the 100% assigned amplicon sequence variants (ASVs), limiting the detection of mismatches and taxa not represented in DNA databases[55]. Therefore, allowing mismatches is essential for investigating both extirpated and potentially extinct taxa. In addition, ecological principles, including the dynamic nature of species distribution ranges influenced by biotic and abiotic factors[56], offer valuable insights. Distinct communities, shaped by significant species interactions, can be identified across space and time[57], potentially including species that have become extinct at specific points of change in community composition among sites. Beta diversity can be used to describe changes in community composition through time and estimate the turnover component of community dissimilarity change representing the contribution of replacement between distinct species, phylogenetic lineages or functional attributes. Therefore, applying a community-based approach on DNA sequences absent from modern DNA samples but evident in past communities may provide clues to potential extinctions.

This study aims to address critical gaps in understanding plant extinctions during the Pleistocene–Holocene transition by analysing ancient DNA metabarcoding data from sediment cores spanning the last 28,000 years, collected from lakes in Siberia and Alaska (Beringia). By integrating established methods with a novel approach for identifying unknown taxa and tracing potential extinction signals, we estimate both local extirpation rates and potential global extinction rates. Our findings reveal that plant extinctions are closely associated with shifts in temperature, vegetation dynamics and megafauna extinctions during this period of environmental upheaval. We estimate extinction rates of 1.7–5.9 extinctions per million species years (E/MSY), higher than background levels but lower than modern estimates. Major extinction events are detected around 17,000 and 9000 years ago, lagging behind peaks in vegetation turnover. Herbaceous taxa and species contributing less to beta diversity are more vulnerable to extinction. While expanding DNA reference libraries and sedaDNA datasets will further refine these estimates, our results suggest that plants demonstrate greater resilience to past environmental changes compared to mammals.

## Results
### Rates of potential plant extinctions
We analysed metabarcoding data from 504 samples from eight lake sediment cores representing seven lakes from northeastern Siberia and Alaska, each covering at least the last 28,000 years (Table 1). To allow for a reliable taxonomic assignment, we set up a customised database, named *SibAla_2023*, for Siberia and Alaska (55–90°N, 50–150°E and 40–90°N, 150°E–140°W) including all vascular plant species from Embryophyta clades Tracheophyta, Bryophyta, Marchantiophyta and Anthocerotophyta which have >10 regional occurrences in the Global Biodiversity Information Facility database (GBIF[58]) and which have available *trn*L P6 loop sequences from public databases (arctborbryo[59–61]; EMBL 143[62]; PhyloNorway[51]). *SibAla_2023* includes 3398 species (representing a 70% GBIF database coverage) and a total of 2371 unique sequence types.

The metabarcoding analyses yielded a total of 128,619,625 reads, which were assigned to 23,005 ASVs. To ensure reliable and real taxa detection, we applied a quality filter in the metabarcoding sequencing results, checking polymerase chain reaction (PCR) replicates' replicability. Assignment was allowed if a sequence showed 90–100% similarity to the reference database to allow for the detection of potential extinct taxa. Counting only the ASVs with a minimum of 100 reads, 332 were assigned with 100% similarity to the reference database and a further 5089 were ASVs with 90–99% similarity (Supplementary Data 1). To exclude sequences that can originate from PCR or

**Table 1 | Summary information on the eight lake sediment cores investigated**

| Core ID | Site—Lake | Latitude (decimal degree) | Longitude (decimal degree) | Collection date (CA) | Length (cm) | Number of samples | Age range (cal. yrs BP) | Age model—references | sedaDNA studies—references |
|---|---|---|---|---|---|---|---|---|---|
| PG1755 | Bilyakh | 65.28N | 126.78E | 2005 | 936 | 57 | 50,000 | 110 | – |
| PG2133 | Bolshoe Toko | 56.25N | 130.5E | 2013 | 380 | 47 | 34,000 | 30 | 30 |
| E5-1A | E5 | 68.63N | 149.45W | 2014 | 500 | 61 | 35,000 | 111 | – |
| Co1412 | Emanda | 65.28N | 135.75'E | 2017 | 610 | 45 | 57,000 | 112 | – |
| EN18208 | Ilirney | 67.35N | 168.32E | 2018 | 1055 | 82 | 52,000 | 113 | – |
| 16-KP-01-LO2 | Ilirney | 67.35N | 168.32E | 2016 | 235 | 58 | 28,000 | 114 | 49 |
| Co1401 | Levinson Lessing | 74.45N | 98.65E | 2017 | 4600 | 88 | 62,000 | 115 | – |
| EN18218 | Rauchuagytgyn | 67.78N | 168.73E | 2016 | 653 | 66 | 29,000 | 116 | – |

sequencing errors, we developed a quality approach accounting for co-occurrence patterns, and statistically detected 10 communities comprising at least five ASVs within the built network (Supplementary Data 2). By collapsing all ASVs which are part of similar communities and with similar taxonomic names, we transferred the ASVs signal into a taxa signal (Supplementary Note 2). The effectiveness of the community approach to identify real taxa was indicated by the fact that for 14 of 16 taxa represented by more than one unique ASV, ASVs assigned to similar taxa are part of the same community (see Supplementary Data 4).

To allow investigation of appearance and disappearance of taxa through time, 2000-year time-slices covering the last 28,000 years were set. A 1000 iteration resampling of number of cores, number of samples and reads was performed to ensure proper comparison between time-slices. After resampling, the final dataset includes 359 taxa (Fig. 1). Among them 216 are dbtaxa i.e. taxa derived from ASVs with similar co-occurrence patterns and 100% match to the *SibAla_2023* database. A total of 143 taxa are non-dbtaxa, i.e. taxa derived from ASVs with similar co-occurrence patterns but only 90–99% match to *SibAla_2023* (Supplementary Data 3 for the median abundance).

Of the 359 taxa, 127 are lost (i.e. absent from the modern time-slice (0–2000 years time bin); Fig. 2a). Among them, 67 are lost dbtaxa (53%) which are absent from the sites in the modern time-slice but still present in the study region as they match 100% with a reference in the *SibAla_2023* database. Thus, based on the dbtaxa only, the loss of dbtaxa at these seven sites is 31%. Furthermore, there are 60 lost non-dbtaxa (47%), i.e. they are absent from the modern time-slice and are not covered by the *SibAla_2023* database. In our dataset, the 60 lost non-dbtaxa represents the maximum number of potentially extinct taxa (Fig. 2c). The loss of non-dbtaxa is 42%, and thus by 11% higher than the lost dbtaxa.

We set up a synthetic dataset to test whether the observed portion of 67/60 of dbtaxa/non dbtaxa missing in the modern time-slice can plausibly only originate from the fraction of GBIF plant species covered/not covered by the *SibAla_2023* (Supplementary Note 2). The synthetic dataset mimics our palaeorecord by resampling GBIF according to the observed taxa gains and losses between consecutive time-slices. For taxa present in the modern time-slice the simulated dataset has a median proportion of 70/30% dbtaxa/non-dbtaxa reflecting the 70% database coverage of GBIF taxa (Fig. 2b). Within error bars, this is in intriguing agreement with the observed proportion of 64/36% dbtaxa/non-dbtaxa observed by our *seda*DNA dataset in the modern time-slice. In contrast, the simulated proportion of 67/33% for db/non-dbtaxa absent in the modern time-slice differ from the observed proportion of 53/47% db/non-dbtaxa absent in the modern *seda*DNA time-slice. Thus, the fraction of non-dbtaxa missing in the modern *seda*DNA time-slices is higher than expected based on database cover. From this result, we conclude that of the 60 lost non-dbtaxa, 42 could be expected from the database coverage and

therefore, a minimum of 18 taxa are likely extirpated from the study region or even potentially globally extinct.

The distribution of rare to common taxa is similar between the GBIF fractions covered and not covered by the *SibAla_2023* database (Supplementary Fig. 2). Therefore, we consider it unlikely that an unequal representation of rare taxa might have artificially created the extinction signal.

We further assessed whether the lost non-dbtaxa are more likely extirpated from the region of Siberia and Alaska or even potentially extinct. We found that 88% (54/60) of the lost non-dbtaxa have no match with other global databases and are therefore potentially extinct globally (Supplementary Note 2.4). For the six taxa that show a 100% match with other databases, we assessed the likelihood that they have modern refugia outside the study region of Siberia and Alaska by assessing the species overlap between regions where the taxa are found and the region of Siberia and Alaska at family level. Our analyses yielded a rather low likelihood for outside refugia of <5% for two taxa and <30% for four taxa (Supplementary Note 2.4) with an average of 18.2% of the six detected taxa, representing one taxon which likely has an outside refugium and is, therefore, extirpated and not extinct. From this analysis, we conclude that 59 of 60 lost non-dbtaxa are potentially extinct.

Taking the results of all analyses together, we conclude that 17 to 59 lost non-dbtaxa are potentially globally extinct. Thus, the potential extinction rate ranges between 1.7 and 5.9 E/MSY over the last 28,000 years in the study area.

## Temporal pattern of potential taxa extinction and correlation to potential drivers

In our dataset, taxa disappeared and/or appeared frequently over past time-slices. We estimated the potential extinction rate with confidence intervals by accounting for the likelihood of a taxa to reappear given the remaining time-slices until present time (Fig. 3a, see Supplementary Note 3 and Supplementary Figs. 5–7).

Potential plant extinctions mainly occurred in two waves around 17,000–15,000 cal. yrs BP (10.6% above the expected rate, with a confidence of 100–96%) and 9000 cal. yrs BP (19.4% above the expected rate and 97% confidence). For comparison, loss of dbtaxa occurred mainly between 19,000 and 9000 cal. yrs BP and with a peak at 13,000 cal. yrs BP which is 26.7% higher than the expected rate and has a confidence of 99.5% (Supplementary Note 3).

A synthetic dataset was created using the *SibAla_2023* database to simulate random ASV loss and the corresponding progression of species loss. This allowed to confirm that the taxonomic resolution of the analysed taxa (assessed through a community approach and not always assigned to the species level) likely did not influence the results and that the non-dbtaxa loss approach accurately reflects the actual pattern of species loss (Supplementary Note 2.5).

Our data indicate a compositional change in vegetation during the Pleistocene–Holocene transition, from predominantly forbs and graminoids (67–78% of the taxa detected) before 15 ka to an increased number

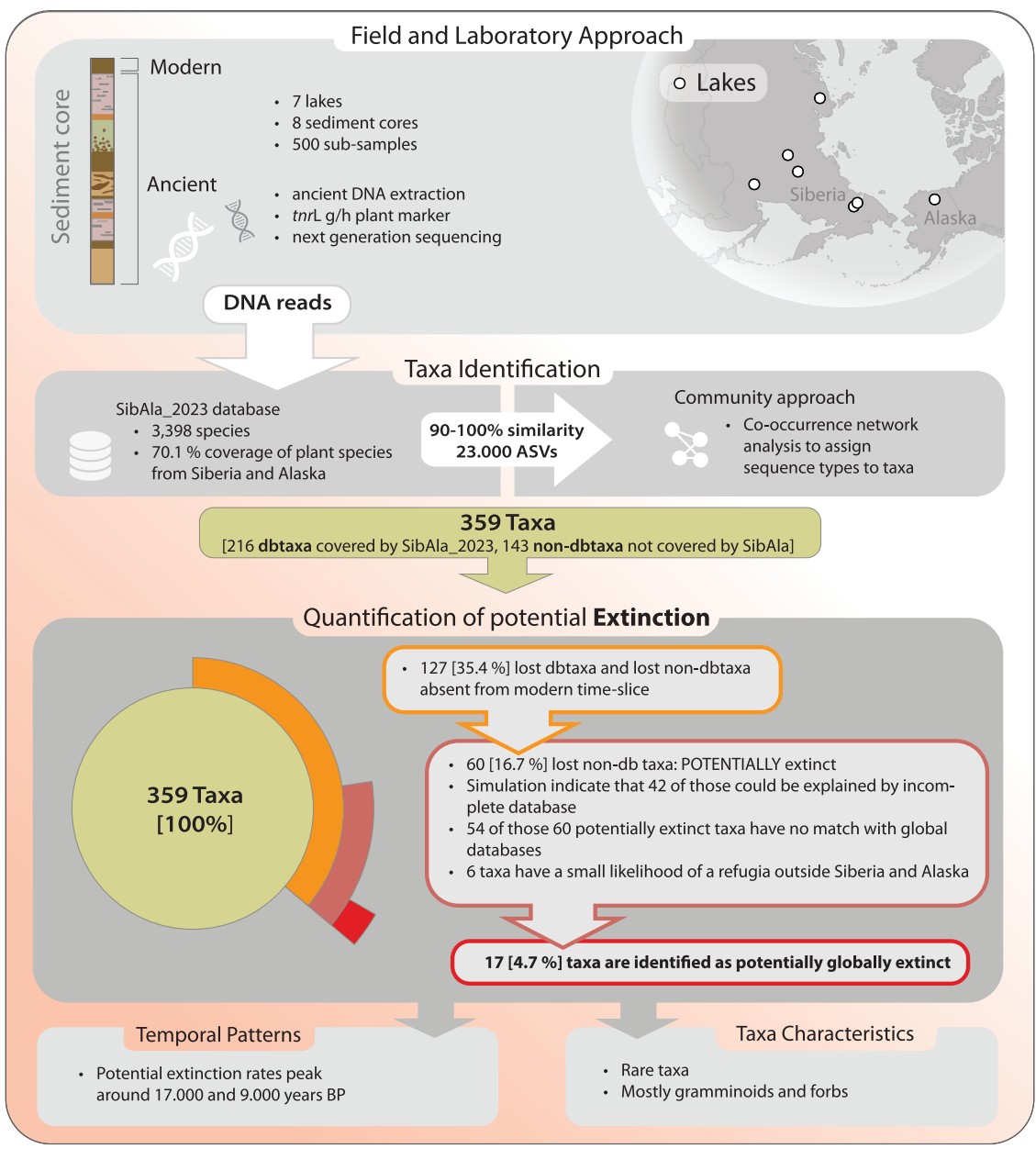

**Fig. 1 | Workflow from data collection, DNA extraction, taxa identification, towards the quantification of extirpation and potential extinction.** We collected eight lake sediment cores from seven sites in Siberia and Alaska. DNA was extracted from subsamples taken at different depths, corresponding to various sediment age layers. After amplifying the *tnr*L P6loop region, amplicon sequence variants (ASVs) were identified and assigned to taxa using the *SibAla_2023* database, with confidence levels ranging from 90% to 100%. To discern true taxonomic signals from potential chimeric sequences caused by PCR or sequencing errors, we developed a co-occurrence-based community detection approach. This analysis identified 359 taxa, with 216 taxa matching the *SibAla_2023* database at 100% confidence (dbtaxa) and 143 taxa matching at 90–99% confidence (non-dbtaxa). From these 359 Taxa, 242 taxa are also present in the modern time-slices and are thus identified as neither extirpated nor extinct. Of these 359 taxa, 242 are also present in modern time-slices, indicating they are neither extirpated nor extinct. Among the remaining 117 taxa, 67 are considered extirpated, while 60 are absent from the region-specific *SibAla_2023* database. These 60 taxa could be missing from the database, extirpated from Siberia and Alaska, or potentially globally extinct. After evaluating database coverage with a synthetic dataset simulated using data from GBIF, we found that 42 taxa could be missing due to insufficient coverage in the *SibAla_2023* reference database. This suggests that 18 (60 minus 42) non-dbtaxa might be globally extinct. Further checks with global databases showed that 54 of the 60 taxa (88%) could not be assigned. Among the six taxa matching global databases, there is an 18% likelihood of having refugia outside Siberia and Alaska, suggesting that one taxon (1 out of 18–60) might only be extirpated. In conclusion, 17 taxa found in sediments from the last 28,000 years are potentially globally extinct. Modified DNA strand Vectorportal.com (https://vectorportal.com/) by CC BY (https://creativecommons.org/licenses/by/4.0/).

of shrubs and trees (22–33%, Supplementary Fig. 3). The assessment of the driver of potential plant extinction rates using a generalised linear mixed effect model (Intercept; Chi2 = 456.9; $p$ = 0.001) highlights vegetation turnover (Fig. 3b) 'shifted by one time-slice' is most important (Chi2 = 30.3; $p$ < 0.001), followed by the megafauna extinction using extinction frames reported in refs. 63,64 (Fig. 3d) (Chi2 = 16; $p$ < 0.001), and July temperature changes (Fig. 3c, Chi2 = 6.8; $p$ = 0.009; see Supplementary Note 4 and Supplementary Table 7).

## Characteristics of the potentially extinct taxa

In this study, 48% dbtaxa are assigned to species level, 38% to genus and 15% to family level. Herbaceous taxa are the dominant functional

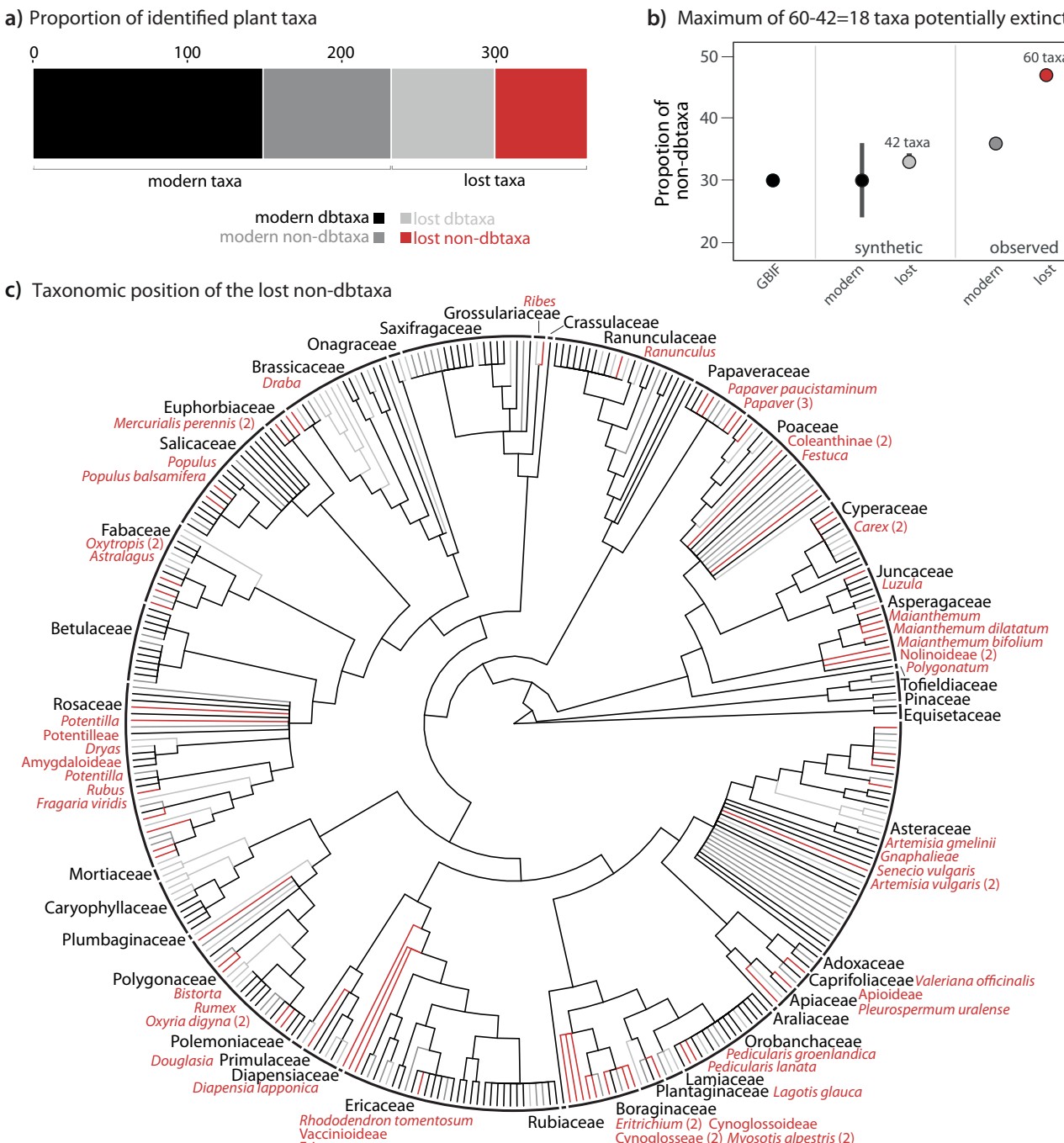

**Fig. 2 | Information about the identified plant taxa. a** Number of taxa assigned modern dbtaxa, modern non-dbtaxa, lost dbtaxa and lost non-dbtaxa. **b** Percentage of taxa with 90–99% match to *SibAla_2023* database (i.e. non.dbtaxa). Taxa with occurrences in the Global Biodiversity Information Facility database (GBIF in black), simulated and observed in present and absent from the modern time-slice. The error bars are the standard deviation for the synthetic data and the dots represent the median values. The observed percentage of taxa in the dataset absent from the modern time-slice and from *SibAla_2023* (i.e. lost non-dbtaxa) is marked in red. The number of lost non-dbtaxa is highlighted both for the observed dataset[60] and for the synthetic[42] as the proportion relative to the total number of 359 taxa observed in the dataset. **c** Taxonomic tree illustrating the number of plant families detected and the taxonomy of the modern dbtaxa, modern non-dbtaxa, lost dbtaxa and lost non-dbtaxa with a number if this taxon is present more than once.

group with 267 assigned taxa (29 graminoids and 238 forbs), followed by shrubs with 66 taxa and trees with 26. Asteraceae, Ericaceae, Rosaceae, Ranunculaceae and Polygonaceae are the most represented families of the 37 detected plant families.

Herbaceous taxa have a higher proportion of lost non-dbtaxa than woody taxa (shrubs and trees, Fig. 4d). On average, 23.3% of the forbs are lost, which is more than the relative proportion of the lost

graminoids (21.6%), lost shrubs (14.1%) and lost trees (10.3%). All differences are significant (*t*-test, *p* value < 0.001).

The lost non-dbtaxa are more likely specialists than the other taxa (Fig. 4a). Lost non-dbtaxa are part of communities with a mean size of 39.85 taxa, which is significantly lower than the community size of other taxa, meaning all taxa that are not lost non-dbtaxa (41.37 taxa; wilcox.test, two-sided, *p* value < 0.001). Lost non-dbtaxa have a low

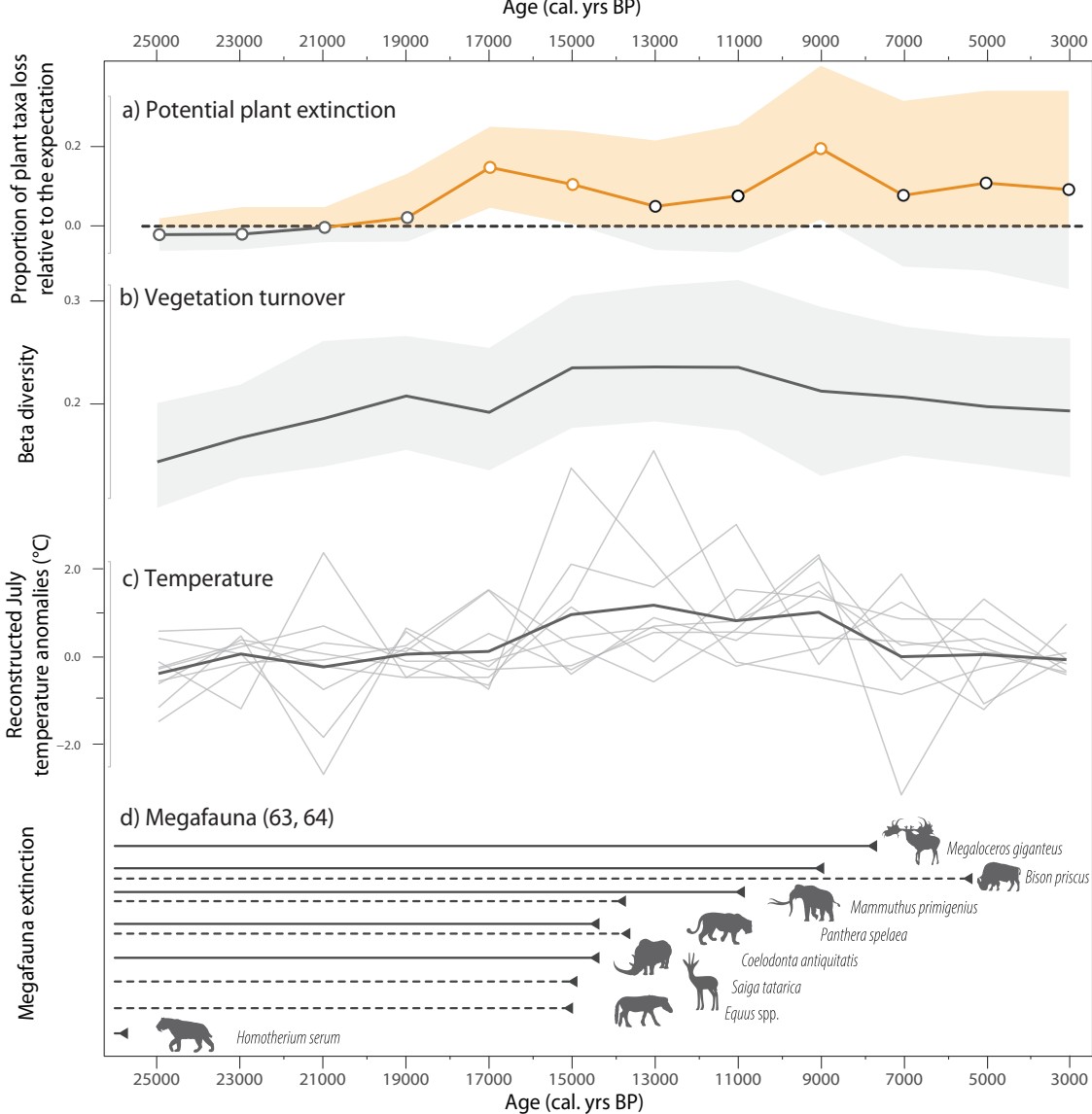

**Fig. 3 | Estimates of potential extinction events in parallel to biotic and abiotic changes over the last 25,000 years. a** Proportional estimates of potential plant extinctions, calculated as the ratio of observed non-dbtaxa losses to expected non-dbtaxa losses. The shaded orange and grey areas indicate the 5th–95th percentile distribution of the observed proportion of lost taxa across 1000 resampling iterations. The line represents the median value of the observed proportion of lost taxa from these iterations. The line is displayed in orange when observed extirpation exceeds the expected baseline (*y* = 0), and orange dots indicate instances where confidence exceeds 80%. **b** Median estimates of vegetation turnover derived from 1000 resampling iterations for both db and non-dbtaxa. The shaded grey area encompasses the 5th–95th percentile range of the iterations. **c** Mean temperature anomalies reconstructed from pollen data (modern analogue technique) over the past 25,000 years, based on analyses from nine sites across the study region (shown as light grey lines). **d** Estimated megafaunal extinction time frames in Alaska/Yukon (dashed lines) and Eurasia (solid lines), sourced from previous research (refs. 63,64, and related literature). All silhouettes were reused from PhyloPic (https://www.phylopic.org/) under CC0 1.0.

species contribution to beta diversity (SCBD) score (6.4e-5), that is, they contribute significantly less to the beta diversity than the other taxa (Fig. 4b; 2.4e-3; wilcox.test, two-sided, *p* value < 0.001). The lost non-dbtaxa are slightly more phylogenetically close to each other (average cophenetic distance of 241.9.39) than to the other taxa (253 for average cophenetic distance between the lost non-dbtaxa and the other taxa; Fig. 4c).

Lost non-dbtaxa are most abundant in the family Boraginaceae (average lost non-dbtaxa in the family relative to the total number of taxa, 55.9%; Fig. 4e). The proportion of lost non-dbtaxa decreases from Asparagaceae (49.4%, *t*-test, two-sided, *p* value < 0.001), through Papaveraceae (49%, *t*-test, two-sided, *p* value = 1), Fabaceae (29.5%, *t*-test, two-sided, *p* value < 0.001), Polygonaceae (25.4%, *t*-test, two-sided, *p* value < 0.001), Rosaceae (24.1%, *t*-test, two-sided, *p* value =

0.02), Poaceae (23.3%, *t*-test, two-sided, *p* value = 1), Juncaceae (20%), Asteraceae (19%, *t*-test, two-sided, *p* value = 0.9), Cyperaceae (18.8%, *t*-test, two-sided, *p* value = 1), Brassicaceae (15.3%, *t*-test, two-sided, *p* value < 0.001), Orobanchaceae (15.1%, *t*-test, two-sided, *p* value = 1), Ericaceae (12.5%, *t*-test, two-sided, *p* value < 0.001), Salicaceae (11.4%, *t*-test, two-sided, *p* value = 0.3), Ranunculaceae (6%, *t*-test, two-sided, *p* value < 0.001), Betulaceae (4.8%, *t*-test, two-sided, *p* value = 0.1), and Saxifragaceae (3.6%, *t*-test, two-sided, *p* value = 0.03) to Pinaceae (1.3%, *t*-test, two-sided, *p* value < 0.001).

## Discussion
Between 28,000 and 2000 cal. yrs BP, 17 (4.7%) to 59 (16.4%) of the 359 assigned taxa were identified as potentially becoming globally extinct. We employed a novel systematic data analysis method designed to

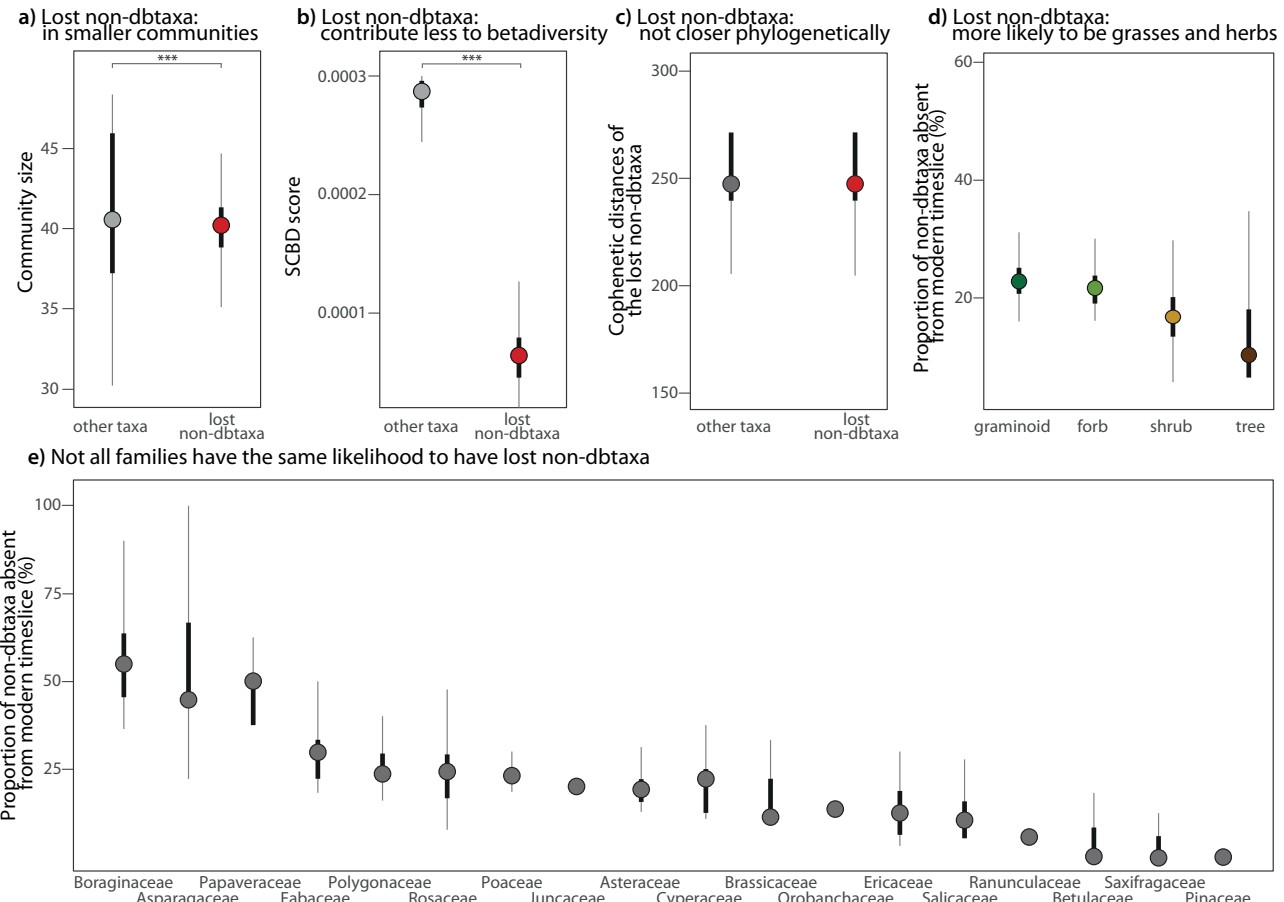

**Fig. 4 | Characterisation of potentially lost plant taxa with the original distribution of the 1000-time resampled data summarised as boxplots.** For all boxplots, the dot represents the median value of the 1000-time resampled data, the black line delimits the 25%–75% percentiles and the light grey line the 5%–95% percentiles. **a** Differences in community size between lost non-dbtaxa and the other taxa (modern non-dbtaxa and modern and lost dbtaxa); difference is significant (Wilcoxon test, two-sided, *p* value < 0.001). Lost non-dbtaxa are part of smaller communities compared to the other taxa. **b** Differences in species contribution to beta diversity (SCBD) score between the lost non-dbtaxa and the other taxa; difference is significant (Wilcoxon test, two-sided, *p* value < 0.001). Lost non-dbtaxa contribute less to beta diversity compared to other taxa. **c** Cophenetic distances between the lost non-dbtaxa and between the lost non-dbtaxa and the other taxa. The difference is small and not significant (Wilcoxon test, two-sided, *p* value = 0.27). Lost non-dbtaxa are not phylogenetically closer together than to other taxa. **d** Proportion of lost non-dbtaxa relative to the number of taxa for each plant functional type; all differences are significant (*t*-test, two-sided, *p* value < 0.001). Lost taxa are more likely to be forbs or graminoids than to be shrubs or trees. **e** Proportion of lost non-dbtaxa relative to the number of taxa for each plant family. Plant families do not have the same likelihood to have lost non-dbtaxa.

calculate the proportion of taxa in our study area that may have faced global extinction during this period. Our study, limited to eight cores from seven available sites, establishes a crucial baseline by identifying and characterising potential extinction candidates, which will guide future research efforts aimed at refining these estimates through more extensive sampling to improve spatial resolution of the past mammoth steppe combined with extending the DNA reference databases. It is essential to note that our assessment identifies these taxa as potential extinction candidates rather than confirming their extinction. For example, future studies may discover one of the taxa we labelled as potentially extinct in fossil records or modern databases. However, new data may also reveal additional taxa that could be extinct. Therefore, our innovative quantitative assessment of potential extinction should be, to some extent, robust against major limitations of our study, such as the large spatial area covered by a limited number of sites and specific taphonomic biases, including the preferential recording of ancient DNA from organisms abundant in the vegetation and growing near the lake[65,66].

We found that most of the lost taxa in the Siberia and Alaska region, referred to as 'lost non-dbtaxa' potentially became globally extinct. Our assessment of global reference databases (Supplementary Note 2) and floral similarities (Supplementary Table 4) indicated a low likelihood—only 1 (18%)—of these lost taxa potentially surviving in refugia outside our study area. While this result aligns with the large and environmentally diverse study area, there is still a possibility that smaller refugia exist in regions underrepresented in global genomic and biodiversity databases. This underscores the need to complete these databases, especially with records from under-sampled global regions. In particular, upland steppe areas in Central Asia and the Middle East have been identified as refugial areas for glacial flora[39]. This situation potentially applies to one of the lost Boraginaceae non-dbtaxa, which might have only been extirpated from the Siberia–Alaska region but still exists as *Eritrichium caucasicum*, a species with a small distribution range in the Caucasus[67].

It is likely that some taxa shifted their distribution northward as woody taxa expanded. This is suggested by the absence of about one-third of the dbtaxa in modern time-slices. In the Polar Ural, a reduction in forbs has been observed during periods of shrub and tree expansion[55]. However, those taxa reappeared in modern layers and/or were found in modern vegetation surveys. In contrast, the forbs in our Beringian study sites did not reappear in the modern time-slices. Additionally, we observed that the loss of non-dbtaxa is

slightly higher than the local loss of dbtaxa, indicating that at least some of the lost non-dbtaxa likely represent extinctions rather than extirpation.

We confirmed that at least 18 of the 60 recorded taxa losses cannot be attributed solely to limitations in the *SibAla_2023* reference database coverage. To verify this, we used a synthetic dataset derived from GBIF information, taking into account the taxonomic resolution of the marker and characteristics from our fossil dataset. We also concluded that the observed taxa losses cannot be explained by rare taxa missing from the database, which is significant because taxa are expected to become rare before going extinct[37]. Overall, we believe that establishing a database tailored to our study region, along with the high coverage of *SibAla_2023*—89.4% at the genus level and 70.1% at the species level—substantially contributed to the robustness of our results.

Identifying potentially extinct taxa by matching sedimentary ancient DNA sequences to reference databases is challenging because these databases are almost exclusively composed of modern extant species. To address this, we lowered the similarity threshold in the sequence-to-reference matching process to retrieve candidate sequences representing real taxa not covered by the database. This adjustment, however, made it difficult to distinguish real sequences from corrupted ones. To overcome this, we developed a novel community-based approach to assess the recorded ASVs. This innovative method allowed us to detect 40% more taxa than would have been identified using the classic approach, which relies solely on ASVs matching 100% to the database. Our observation that ASVs from taxa represented by multiple ASVs were usually assigned to the same community confirms the reliability of our approach. Using the detection of a common spatiotemporal pattern as a criterion for taxa identification is considered conservative and may risk overlooking taxa that are part of less widely distributed habitats. By collapsing non-database ASVs with database taxa within the same community, we excluded ASVs likely originating from PCR or sequencing errors. However, this conservative approach may overlook the extinction of taxa with similar ecological requirements (i.e. similar co-occurrence patterns) to closely related surviving taxa, especially for those taxa that are poorly resolved by the marker. Additionally, the potential for hidden diversity, including lost taxa, remains high for ASVs assigned at 100% similarity but not to the species level (113 taxa in this study), such as those in the Rosaceae or Salicaceae families.

Genetic markers that perform best with ancient and short DNA fragments are obviously limited and suboptimal for the discrimination of species[68]. For example, taxa from the Boraginaceae family are difficult to detect in DNA analyses, because of the synthesis and storage by most Boraginaceae genera of pyrrolizidine alkaloids, which may cause rapid and permanent DNA damage[69,70]. Therefore, Boraginaceae is less represented in databases and our study may overestimate the non-dbtaxa assigned to Boraginaceae. Further, Asparagaceae (especially *Convallaria*) is a known contaminant in the g/h datasets that could originate from the DNA extraction kits[30]. However, the potentially extinct Asparagaceae detected in this study did not match either *SibAla_2023* or other databases. Therefore, the *Convallaria* signal detected in other studies could derive from lost Asparagaceae absent from the databases. Recently developed *seda*DNA capture approaches may provide an opportunity to reach better taxonomic resolution, as exemplified by recent studies on capturing the whole *Larix* chloroplast genome[22,71] from lake sediments while other studies have yielded a limited average taxonomic resolution so far[72]. Finally, in parallel to *seda*DNA approaches, further efforts on characterising macrofossils in tandem with aDNA sequencing could help confirm and identify extinct plant taxa signal from past records.

## Potential plant extinction rate

Based on our inferred minimum of 17 and maximum of 59 potentially extinct taxa, the plant extinction rate over the last 28,000 years can be estimated to range between 1.7 and 5.9 potential extinctions per million species years (E/MSY). Thus, the numbers are 5 and up to 116 times higher than the overall background extinction rate of plants (0.05–0.35 E/MSY[12]), yet lower than the estimated currently ongoing global plant extinction rate of 18–26 E/MSY[17]. Comparison of the fossil-to-modern extinction rate should be treated with caution since background rates can be highly taxa specific ([1] and references therein) and are potentially biased towards taxa that are easier to detect within the fossil record. Also, our estimates are lower than the rate of loss of db-taxa 6.7 E/MSY. Taking into account that some of the lost non-dbtaxa may be due to extirpation, and also the low detectability of rare taxa in metabarcoding studies[65], our estimated extinction rates are low.

The measured extinction rate of megafaunal genera over the last 50,000 years, which is only a fraction of the Mammalia, is 12.93 E/MGY (97 out of 150 became extinct[73]). Extinction rates of species and genera should be broadly similar[5]. We, therefore, conclude that the observed potential plant extinction rate at the transition between the Pleistocene and the Holocene is, at most, 2.2–7.6 times lower than that of megafauna[63,73]. According to modern estimations, mammalian extinction rates are much higher (up to 1000 times above the expected background extinction[3]) than for plants (up to 500 times above the background extinction rates[17]). Hence, our evidence for taxa loss on multi-millennial time-scales supports results from modern ecological data tracing taxa loss at a multi-decadal scale, which suggests that, compared to other taxonomic groups, plants are more resilient to extinction, likely because they possess mechanisms of dieback (e.g. seed dormancy[74]), may have high longevity[75] or can reproduce vegetatively[76].

## Plant extinction dynamics and drivers

During the transition from the late Pleistocene to the Holocene, potential plant extinctions occurred between 19,000 and 9000 calibrated years before present (cal. yrs BP, Fig. 3a). Specifically, two peaks of potential plant extinction were identified within this timeframe: one around 17,000 cal. yrs BP, and another around 9000 cal. yrs BP. Our correlation analysis identified vegetation turnover with a lag of one time-slice as the most important driver (Fig. 3b), followed by the megafauna extinction (Fig. 3d) and July temperature changes (Fig. 3c).

A major part of the plant species pool was dynamic over the last 28,000 years, with only 35% of detected plant taxa present in all 2000-year time-slices. We found highest compositional turnover at the Pleistocene–Holocene transition confirming plant compositional changes at the Pleistocene-Holocene transition reported by other studies from Siberia and Alaska[30,59]. During the Pleistocene, graminoids and forbs were the dominant species. However, at the beginning of the Holocene, these species were replaced by woody taxa. The observed time lag between species turnover and potential plant extinction might be explained by the loss of previous interactions within the species pool. Some plant taxa, unable to adapt to these changes or track range shifts, can eventually go extinct[77].

At the end of the Pleistocene, megafaunal extinctions occurred in waves in the Holarctic, with the first wave during the last glacial maximum (LGM) and the second between 15,000 and 10,000 cal. yrs BP[22,64]. In addition, two pulses of community turnover of smaller mammals have been detected around 14,500 cal. yrs BP and between 11,000 and 7500 cal. yrs BP, with evidence for extirpations in North West America[78]. These events align closely with the observed two peak periods of potential plant extinction (Fig. 3). This suggests a strong relationship between megafaunal extinction and potential plant extinction during the Pleistocene-Holocene transition. The now extinct Pleistocene megaherbivores, which primarily fed on forbs and grasses, played a crucial role in maintaining plant diversity and shaping

their environment[30,59]. As megafauna disappeared, habitat fragmentation likely occurred, affecting plant communities[79,80]. This fragmentation, combined with rapid and repeated climate shifts characteristic of the Pleistocene–Holocene transition, may have contributed to the potential extinction of plant species.

No direct correlation was identified between the potential extinction events and warmer or cooler climates (Fig. 3), instead, major potential plant extinctions are influenced by environmental instability, tracking changes in climate rather than specific temperature levels. This provides fresh arguments from consistent time-series data that environmental stability plays a significant role in determining the diversity gradient across latitudes, with higher latitudes experiencing greater instability compared to the tropics[81].

This study highlights the occurrence of potential plant extinctions during the Pleistocene–Holocene transition in Siberia and North West America, correlating with abrupt shifts in temperature, megafauna extinction and plant species turnover with a time lag. While the Pleistocene was characterised by millennial-scale climate fluctuations[82] with no record of major extinction events (of mammals or plants), the transition to the Holocene witnessed extinctions of Pleistocene megafauna, likely contributing to changes in plant communities[21,31,83]. In our study, plant compositional change (with time lag) rather than megafauna extinction yielded the highest correlation with plant extinction dynamics in line with recent studies which call the keystone role of megafauna in shaping Pleistocene grasslands into question[26,31]. However, our study was not designed to disentangle these temporally correlated drivers. Further research, including modelling and high taxonomic and time resolution sedaDNA data[84], is needed to fully discern the drivers of plant extinctions during this critical transition period.

### Characteristics with higher vulnerability to plant extinction

Potentially extinct plant species during the Pleistocene–Holocene transition contributed less than average to ecosystem functionality, as indicated by their SCBD. This finding aligns with contemporary observations of plant extinctions[37]. This suggests that the disappearance of the mammoth steppe biota was not primarily driven by the extinction of key plant species, as the lost taxa were neither abundant nor critical contributors to the ecosystem's functioning.

We also found that potentially extinct plant species were more likely specialists than generalists. The transition from the Pleistocene to the Holocene witnessed rapid alterations in vegetation structure, leading to the emergence of new ecological assemblages such as tundra and taiga[24,39]. Generalist plant species, which exhibit greater adaptability to environmental changes, were more likely to persist in these new ecological settings[85–87]. Conversely, taxa with narrow ecological niches were more vulnerable to habitat changes and were consequently potentially more susceptible to extinction, a pattern observed in previous studies[88,89]. However, exceptions exist, such as instances where abundant species undergo extinction (i.e. ref. 90), or where taxa persist as rare entities without facing extinction (i.e. ref. 91), highlighting the complexity of extinction dynamics.

Our phylogenetic analysis revealed that the potentially extinct plant taxa were only minimally more closely related to each other than to other taxa, indicating that potential extinction events were not confined to specific taxonomic groups. This confirms modern observations[37] as extinction across various phyla can occur at the species level (i.e. shotgun extinction principle[42]) making detection of past taxa to species level of high importance for studying past extinction. Nonetheless, certain plant families, such as Boraginaceae, Papaveraceae and Polygonaceae, exhibited higher likelihoods of potential extinction. For example, Boraginaceae *Eritrichium* reads are prevalent in Pleistocene datasets from Siberia but less in the Holocene, suggesting their susceptibility to extinction during the transition period[49,92]. Similarly, extreme cooling events at the end of the LGM[93] potentially have driven the extinction of Polygonaceae taxa that

reached their temperature limit as most are distributed in temperate regions[94]. Additionally, Papaveraceae taxa, dominant in Pleistocene sedaDNA records[95], might have succumbed to competitive pressures following the invasion of more competitive woody taxa.

A greater potential for extinction was detected among forbs and graminoids compared to shrubs and trees following the loss of the mammoth steppe. This ecosystem was characterised by a rich diversity of plants, primarily forbs and grasses[31,32], which were then most affected by the Pleistocene-Holocene transition loss of species richness and diversity[30,49]. During the LGM, plant range shifts were observed, and tree and shrub species survived in refugia before recolonising during the Holocene[96,97].

### Outlook

We leveraged the P6loop of *trn*L (UAA) intron plant marker on sedaDNA from eight sediment cores to assess potential plant extinctions in the course of the Pleistocene mammoth steppe loss in Siberia and North America during the last 28,000 years. We developed and applied a systematic data analysis to identify potentially extinct taxa that can be implemented by the community in further studies involving new locations spanning the mammoth steppe area. Our novel approach includes the setup of a customised reference database, the usage of a low read-to-reference matching threshold, the implementation of a community approach to separate real taxa from chimeric or false positive sequences, the setup of a synthetic dataset sampled from GBIF to assess the potential impact of missing taxa in the genetic reference database and the evaluation of taxa survival outside the study area to prove potential global extinction.

In total, we detected 359 plant taxa i.e. 40% more compared to the traditional approach. We inferred that, potentially, 17 to 59 taxa went extinct. Our inferred potential plant extinction rate is much higher than the previously reported long-term background extinction rate but lower than present extinction rate. This confirms the view of the uniqueness of the present extinction event from a Quaternary flora perspective.

We found that potential extinctions likely mainly occurred during the environmental transition from the late Pleistocene to the Holocene peaking at 17,000 and 9000 cal. yrs BP which lags to major vegetation turnover times on millennial time-scales. By analogy to our results from the past, this may indicate that an extinction response to recent widespread anthropogenic vegetation turnover will happen far into the future.

Finally, we characterised the potentially extinct plant taxa as rare with a low contribution to the beta diversity of the mammoth steppe ecosystem. No phylogenetic pattern was identified, but some families were especially affected by potential extinction such as Boraginaceae and Asteraceae. Graminoids and forbs, which are major contributors to the lost mammoth steppe plant diversity, were potentially more sensitive to extinction. This aligns with findings from modern plant extinction studies[37,98].

Overall, this study underscores the potential of sedaDNA to investigate historical extinction events. The initial estimates provided here will improve as the paleoecology community continues to expand DNA reference libraries and ancient sedaDNA data. Our findings suggest that plants exhibit greater resilience to environmental changes compared to mammals. This research paves the way for a deeper understanding of the interactions between rapid ecosystem loss, plant compositional shifts and cascading extinctions across different taxonomic kingdoms, even in the Arctic, crucial in the ongoing context of climate change.

## Methods

### Site description and timeframe

We used eight lake sediment cores from seven sites spanning the Pleistocene to Holocene transition. The sites are distributed across

Siberia and Alaska covering the Beringia and allowing investigation of vegetation changes over an area previously covered by the Pleistocene mammoth steppe (Fig. 1).

Lake Bilyakh (65°17′N, 126°47′E; 340 m above sea level (a.s.l.)) is situated in the western part of the Verkhoyansk Mountains, about 140 km south of the Arctic Circle. The lake covers an area of ~23 km², has an average water depth of 8 m, and a maximum measured depth of about 25 m. It is fed by direct precipitation as well as several small creeks and streams. The climate is characterised by mean January temperatures around −40 °C, mean July temperatures about 15–19 °C, and annual precipitation of 300–400 mm. Today the plant is dominated by open mountain deciduous forests and woodlands with *Larix*.

Lake Bolshoe Toko (56°05′N, 130°90′E, 903 m a.s.l.) is located in a depression of tectonic and glacial origin, on the northern flank of the eastern Stanovoy Mountain range, it is bordered by moraines of three glacials at its northeastern margins. It is 15.4 km long and 7.4 km wide, with a maximum water depth of about 80 m and surface area of 82.6 km². Mean annual air temperature in the study region is 11.2 °C, ranging from −65 °C in January to +34 °C in July. Annual precipitation varies from 276 to 579 mm. Soil cover is thin and contains large amounts of gravel. Northern taiga dominates the study area, forests consist of *Larix*, *Picea* and *Pinus*.

Lake E5 (68.641667° N, 149.457706° W, 795 m a.s.l.) is situated in the northern foothills of the Brooks Range, northern Alaska, it is located on an older glacial landscape (>125 ka BP), referred to as the Sagavanirktok, and did not glaciated during the LGM. It is a small lake, with ~650 m long × 310 m wide at its widest point, and surface area 0.1 km². The Sagavanirktok River moraine in the Lake E5 basin is almost entirely vegetated with tussock tundra, but along the ridgetop southwest of the lake, a few large erratic boulders protrude 1–2 m above the vegetated surface.

Lake Emanda (65°17′N, 135°45′E, 671 m a.s.l.) is situated in the Yana Highlands east of the Verkhoyansk Mountains in Siberia. The lake is 7.5 km long, 6.5 km wide, has a surface area of 33.1 km² and a maximum water depth of 15 m. The climate is categorised as continental sub-arctic, with large seasonal temperature gradients and a mean annual precipitation of 233 mm. The present-day winter climate is predominantly controlled by the intensity of the Siberian High-Pressure system, which causes very cold and long winter periods with thin snow cover and mean January air temperatures of −44.7 °C. The summer climate is mainly controlled by the Asiatic Thermal Low-Pressure system and a high-pressure system over the North Pacific, resulting in short but relatively warm summer periods with mean July temperatures of 13.0 °C and higher precipitation of 40–50 mm per month between June and August. The soil around the lake is pervaded by permafrost. The modern vegetation is represented by sparse mountain larch (*Larix cajanderi*) forests of lichen (*Flavocetraria cucullata*, *Cladina arbuscula*) type with *Betula exilis* and *Pinus pumila* and of *Vaccinium*-moss and lichen type (*V. vitis-idaea*, *V. uliginosum*, *Aulacomnium turgidum*, *Sphagnum spp.*, *Flavocetraria cucullata*).

Lake Ilirney (67°21′N, 168°19′E, 1790 m a.s.l.) is situated in the region of Chukotka, far east Russia. The lake is bounded by the Anadyr Mountains to the north. The basin itself is 12 km in length and 3.6 km across at its widest point and oriented northeast to southwest along a large structural feature. The basin can be split into two sub-basins: a larger depocenter in the southwest and a smaller basin in the northeast divided by a roughly north-south oriented bathymetric high. The maximum water depth in the southwestern basin and in the northeastern basin is 44 and 25 m, respectively. The catchment area is 1214 km². A strongly continental climate characterises Ilirney with a mean annual temperature of −13.5 °C and average January and July temperatures of −33.4 °C and 12.1 °C, respectively. Regional snowfall is equivalent to 110 mm of water and June–September precipitation totals 70 mm. The lake zone is covered by continuous permafrost and located at the latitudinal tundra-taiga ecotone boundary. The direct vicinity is dominated by open mountain *Larix* forests.

Age models for the sediment records were adopted from previous studies listed in Table 1. The cores cover the last 62,000 years for the oldest core and the last 28,000 years for the most recent one. Therefore, we investigated the last 28,000 years only and split the records into 14 time-slices of 2000 years.

## Sampling and sediment DNA lab work

For each of the lake sediment cores, the opening and subsampling were performed in the climate chamber of the Helmholtz Centre Potsdam−German Research Centre for Geosciences (GFZ) at 4 °C, physically remote from any molecular genetic labs preventing contamination with modern DNA. Precautions for clean subsampling were applied during sediment core subsampling. All surfaces in the climate chamber at GFZ (Potsdam, Germany) were cleaned with DNA Exitus PlusTM (VWR, Germany) and demineralised water before working on the core. The sampling tools such as knives, scalpels and their holders were cleaned before the taking of each sample (as recommended in ref. 99). About 3 mm of each sample slice that was in touch with the plastic tube or the thin foil that covered the half-core was removed using a sterile scalpel as these parts cannot be considered sterile[100]. Collected sediment samples were stored in 8 mL sterile tubes (Sarstedt) at −20 °C.

Molecular genetic analyses of the sediment samples were conducted in the palaeogenetic laboratories of the Alfred Wegener Institute Helmholtz Centre for Polar and Marine Research (AWI) in Potsdam, Germany (as described in ref. 30). This laboratory is dedicated to ancient DNA isolation and PCR setup and is located in a building devoid of any molecular genetic work. It is cleaned frequently and subjected to nightly UV-irradiation. All laboratory work was performed in a UVC/T-M-AR DNA/RNA cleaner-box (BIOSAN, Latvia). DNA isolations and PCR setups were performed on different days using dedicated sets of pipettes and equipment. Further precautions to reduce contamination included UV-irradiation of 10× buffer, BSA, MgSO4 and DEPC-treated water for 10 min in a UV crosslinker in thin-walled PCR reaction tubes (recommendations of ref. 99). Each laboratory step including subsampling, extraction, PCR amplification, purification and pooling, as well as next generation sequencing was done for each core separately. The same laboratory protocols for DNA isolation and PCR were followed for all cores. DNA isolations were performed, each with nine samples and one control (blank). Total DNA was isolated from ~3–5 g of sample material using the PowerMax® Soil DNA Isolation Kit (Qiagen, Germany, cat#12988-10) added to 1.2 mL of C1 buffer, 0.4 mL of 2 mg/mL Proteinase K (protK, Karl Roth, Germany, cat#7528.1) and 0.5 mL of 1 M dithiothreitol (DTT, Omnilab, Germany, cat#1198463). This mixture was homogenised for 10 min on a vibratory mixer (VortexGenie2, Mo Bio Laboratories, USA) and incubated overnight at 56 °C while rotating. All subsequent steps were performed according to the instructions of the manufacturer Qiagen, except for the final elution volume which varied between 1.6 and 2.0 mL. For most core samples, 800 μL of elution buffer (C6 buffer) was applied to the filter membrane, incubated for 10 min at room temperature and then centrifuged for 3 min at 2500 × g. This step was performed twice ending up with a final volume of 1.6 mL. One exception was for the Lake Ilirney (EN18208) core where 2 mL of elution buffer (C6 buffer) was directly added to the filter membrane. For all core sediment samples, except for samples from Lake Bolshoe Toko, 1 mL of the DNA extracts were concentrated using the GeneJET PCR Purification Kit (Thermo Fisher Scientific, cat#K0692) using 100–50 μL elution buffer. The concentrated DNA was measured using the ds-DNA BR Assay Kits ((Invitrogen, USA, cat#Q32853) and the Qubit® 2.0 fluorometer Qubit 2.0 fluorometer (Invitrogen, cat#Q32866). Then, the DNA solution was diluted to 3 ng/μL for the PCR reactions.

The PCR reactions were performed using the trnL g and h primers[68]. Both primers were modified on the 5′ end by unique 8 bp tags which varied from each other in at least five base pairs to distinguish samples after sequencing and were additionally elongated by NNN tagging to improve cluster detection on the sequencing platform. The PCR reactions contained 1.25U Platinum® Taq High Fidelity DNA Polymerase (Invitrogen, USA, cat# 10675783), 1× HiFi buffer (cat#10675783), 2 m MMgSO4 (cat#10675783), 0.25 mM mixed dNTPs (Life Technology, cat#R1121), 0.8 mg Bovine Serum Albumin (VWR, Germany, cat#11761797), 0.2 mM of each primer (IDT) and 3 µL DNA in a final volume of 25 µL. PCRs were carried out in a Professional Basic Thermocycler (Biometra, Germany) with initial denaturation at 94 °C for 5 min, followed by 50 cycles of 94 °C for 30 s, 50 °C for 30 s, 68 °C for 30 s and a final extension at 72 °C for 10 min. The extraction blank and one no template control (NTC) were included in each PCR to identify possible contamination during extraction and PCR set-up. For each extraction sample, three PCR replicates with differently tagged primers were performed. Except for the Bolshoe Toko core, replicates were amplified with the same tag combination. PCR success was checked with gel-electrophoresis on 2% agarose gels (Carl Roth GmbH & Co. KG, Germany). The PCR products were purified using the MinElute PCR Purification Kit (Qiagen, Germany, cat# 28006), following the manufacturer's recommendations, and finally eluted in 20 µL of elution buffer. The DNA concentrations were quantified with the dsDNA BR Assay (Invitrogen, USA, cat#Q32853) using the Qubit 2.0 fluorometer (Invitrogen, cat#Q32866). To avoid bias based on differences in DNA concentration between samples, all replicates were pooled in equimolar concentrations for each core. For each core, all extraction blanks and NTCs were included in the sequencing run, even though they were negative in the PCRs. Each sequencing run was performed by Fasteris SA sequencing service (Switzerland) with the paired-end sequencing on a HiSeq or NextSeq Illumina platform (Supplementary Table 1).

## Data analysis

Each core metabarcoding dataset was analysed individually using OBITOOLS (version 3[101]). Overlapping paired reads were merged using the command *obi alignpairedend*. The unmerged reads were removed with *obi grep -a mode:alignment*. The command *obi ngsfilter* assigned each sequence to its corresponding sample according to the tag combination. Reads were de replicated with the command *obi uniq --merge sample* and cleaned from PCR and sequencing errors using *obi clean -s MERGED_sample -r 0.05 -H*, which keeps head sequences (-H) that lack variants with a count greater than 5% of their own count (-r 0.05). Taxonomic assignments of the ASVs were performed using *obi ecotag* with the use of the customised *SibAla_2023* database.

The customised database *SibAla_2023* has been compiled using different R packages with the following steps. 1. Taxa selection from a given region (55–90°N, 50–150°E and 40–90°N, 150°E–140°W) and taxa occurrences (>10) using the Global Biodiversity Information Facility (GBIF accessed the 16.03.2023[58]) resulting in 233 families, 1059 genera and 4849 species. 2. Alignment between selected taxa and available P6 loop sequences from public databases (arctborbryo[59–61]; EMBL 143[62]; PhyloNorway[51]). 3. Quality filtering of selected P6 loop sequences. 4. Preparation for usage with obi ecotag. The *SibAla_2023* database compared with given occurrences in GBIF amounts to 95.7% (family level), 89.4% (genus level) and 70.1% (species level) of taxonomic coverage. Finally, the *SibAla_2023* database includes a total of 4939 entries, comprising 3398 species, 947 genera and 223 families that collapse into 2371 unique P6 loop sequence types. The entries of the database are encoded with a unique identifier.

Data quality was assessed by investigating the replicability of the PCR replicates and their compositional patterns by non-metric multi-dimensional scaling analyses. The analysis was done core-wise and only ASVs that have a 100% identity match to the reference database were used. The ASVs compositional data were Hellinger transformed with decostand() and ordination was performed with metaMDS() in the R package vegan (version 2.6-2[102]). If replicates of the same sample did not form a cluster but show that one replicate is distant from the others (indicating a different compositional signal), we checked read count and composition of the replicate. If the total read count of the affected replicate was below 100 counts and/or the sample was composed of fewer (less than three) and different plant sequence types as compared to the remaining replicates of the sample, replicates were excluded from the dataset. After replicate evaluation, the replicate removal was done on the raw data (Obitools3 output). The remaining high-quality replicates were merged into samples by summarising the replicate read counts.

The single-core data was filtered for ASVs that have at least a 90% similarity with reference sequences *SibAla_2023*. Finally, the eight single-core datasets were merged into one dataset by keeping only ASVs appearing in at least ten samples with a minimum read count of 100 reads per ASV. From this final dataset, we grouped ASVs with 100% similarity into database ASVs (dbASVs) and the remaining ASVs (with 99–90% similarity) into non-dbASVs.

Given the strong connection between the occurrence of species within plant communities over time and space, we utilised co-occurrence patterns to group ASVs into these communities. This helps enhance our confidence in the taxonomic signals. First, we ran a correlation analysis using the corr.test function in R (Spearman method and the Holm adjustment for non-normally distributed data). We retained only results with a positive correlation score above 0.4 and constructed a network using the graph_from_data_frame function. Next, we used the cluster Louvain function from the igraph R package to detect community structures and optimise modularity scores. Within these identified plant communities containing at least five ASVs, we differentiated between database ASVs (dbASVs) and non-database ASVs (non-dbASVs), assigning them to unique taxa accordingly. To minimise potential errors, we collapsed non-dbASVs to dbASVs within communities if their taxonomic assignments matched. For example, if within the same community, non-dbASVs that are assigned to the same family as dbASVs will be collapsed with the name of the dbASVs under a single dbtaxon. This step helps mitigate the risk of chimeric sequences and other PCR/sequencing artefacts.

We split the lake sediment records into 14 time-slices of 2000 years. The mean age is used: the oldest time-slice covering 28,000–26,000 cal. yrs BP and the most recent and modern time-slice covering 2000–0 cal. yrs BP. For setting up our reads per taxon per time-slice dataset, we performed a three-step resampling script to balance the number of reads, number of samples and number of sites (see Supplementary Note 1 and Supplementary Table 2 and Supplementary Fig. 1). The resampling was performed 1000 times. The results presented are based on the median values obtained from analysing the 1000 resampled datasets. We assessed the occurrence of each dbtaxon and non-dbtaxon in each time-slice. We defined the lost dbtaxa and lost non-dbtaxa as those absent from the modern time-slice.

To test whether the observed portion of lost dbtaxa/non-dbtaxa can plausibly only originate from the fraction of GBIF plant species covered/not covered by the *SibAla_2023*, we created a synthetic dataset to mimic our palaeorecord by resampling GBIF according to the observed taxa gains and losses between consecutive time-slices. For example, if eight and six dbtaxa and non-dbtaxa, respectively were lost between time-slice 27,000 and 25,000, and three dbtaxa and five non-database taxa were gained over the same time period, we randomly removed eight and added three taxa that were in the reference database whereas we removed six and added three taxa that were lacking in the reference database. We then compared the observed dbtaxa/non-dbtaxa ratios to the simulated values. In addition, we compared the distribution of rare taxa in both GBIF and our dataset to check whether

an unequal representation of rare taxa might artificially create an extinction signal.

Finally, to assess the potential presence of lost non-dbtaxa beyond the region used to build the *SibAla_2023* database, we conducted a MegaBLAST[103] against the global, NCBI nt database[104] using Geneious Prime 2023.2.1. This resulted in the assignment of seven ASVs to species level with 100% confidence. For each species assigned, we obtained its occurrence region using GBIF, encompassing all occurrences of the species. Subsequently, we identified the families to which the assigned species belong and compared all species within these families in the study area to those in the region of Siberia and Alaska belonging to the same family. If the number of species exceeded 100, a random selection of 100 species was made from each region for comparison; if fewer than 100 species were present, all were used. This analysis helped determine the overlap of species between the two regions.

We consider the number of all non-dbtaxa absent from modern time-slice and are not covered by the *SibAla_2023* database as the maximum number of potentially globally extinct taxa. The minimum number of potentially globally extinct taxa was retrieved taking into account the analyses of GBIF plant species coverage of *SibAla_2023* and considering lost taxa presence outside the study region of Siberia and Alaska. Accordingly, we estimated the minimum and maximum rate of potential taxa extinctions presented in units of the number of extinctions per million species years (E/MSY; as described in ref. [105]).

Disappearance of taxa in the dataset happens at every time-slice in parallel to the reappearance of taxa. There can be several time-slices between a taxon's disappearance and its reappearance in the dataset. Therefore, potential extinction rates would increase towards the recent time-slices as fewer time-slices are left for a disappeared taxon to reappear in the dataset. For this reason, reappearance rates are the key to identifying potential extinction events.

To establish an expected extirpation rate for each taxon disappearing from a time-slice, we report the number of time-slices it took to reappear in the dataset. Measured from each of the 1000 resampling iterations, a maximum of nine time-slices can be considered for a taxon to reappear once it disappeared. Therefore, we measured the expected extirpation rate using the time-slices ~25,000, ~23,000 and ~21,000 yrs BP, as this leaves nine time-slices for taxa to reappear later in the dataset once they disappeared from a time-slice. At every resampling iteration, we estimated the number of time-slices needed for lost taxa to reappear for each time-slice they disappeared from. We used this distribution and the median number of time-slices needed for taxa to reappear as the expected reappearance rate for each time-slice. We compared this expected reappearance rate to the observed rate at each time-slice for each of the 1000 iteration steps of our resampling. We could then infer a likelihood of extirpation (lost dbtaxa) and potential extinction (lost non-dbtaxa) based on the proportion that the observed reappearance rate is lower than expected for each time-slice (see Supplementary Note 3 and Supplementary Fig. 5). In addition, we could infer the net proportion of taxa loss as the difference between the percentage of observed taxa lost and the expected loss at each time-slice.

To examine whether ASV loss and species loss share the same patterns, we analysed data from the *SibAla_2023* database, which contains corresponding information for ASVs and species. Each ASV may be linked to multiple species, and vice versa, a single species may correspond to multiple ASVs. We simulated ASV loss progress by setting the same ASV loss and reappearance numbers obtained from sedimentary ancient DNA results. Leveraging the ASV species correlation from the *SibAla_2023* database, we simulated species loss progress by matching observed ASV losses in each artificial time-slice. For each time-slice, we determined the species loss and reappearance

numbers. Subsequently, using the same methodology applied to calculate past taxa losses, we calculated the proportion of lost species relative to the expected values. Our results aligned with sedimentary DNA estimates, revealing a consistent pattern. Specifically, periods characterised by substantial taxa loss also exhibited significant species loss.

Generalised linear models (GLM) were built to identify potential correlations between the observed plant loss (extirpation and potential extinction) rate for each of the 1000 iteration steps and biotic and abiotic factors. Only positive plant taxa-loss values were used, and the negative values were set to 0 in order to build a binomial GLM with positive values of plant taxa-loss rate as success (observed plant taxa loss) and negative values as non-success (non-observed plant taxa loss). After fitting with a step for the best likelihood ratio test, the parameters explaining the most variance of the plant extinction were selected (see Supplementary Note 4 and Supplementary Table 7). The factor representing beta-diversity changes that explain the most variance is the replacement rate (vegetation turnover) with a 2000 yrs shift (i.e. an increase in replacement rate occurs one time-slice before the plant taxa loss). The factor that explains the most plant taxa-loss variance for climate change is the pollen-based reconstructed changes in median temperature anomalies using the modern analogue technique[44,106]. For the beta-diversity estimate, the temperature changes and the megafauna extinction time frames (estimated from refs. [63],[64]), a pairwise GLM with binomial distribution was performed against the plant extirpation and potential extinction rates.

To characterise the lost non-dbtaxa, we compared their characteristics with other taxa (and to the taxa present in the modern time-slice; Supplementary Note 5). For each dbtaxon and non dbtaxon, we estimated whether they are more specialist or generalist based on the number of taxa present in their detected communities. We assumed that taxa present in communities with fewer total taxa are more specialist than taxa present in communities with more total taxa. We compared the number of taxa per community for the lost non-dbtaxa and the modern taxa with a Wilcoxon statistic test. Before going extinct, a taxon is likely to become rare and functionally extinct. To estimate the contribution of the taxa to the functionality of the system, we assessed their SCBD scores with the beta.div function from the adespatial R package (version 0.3-14[107]) on square-root relative abundance data. We compared the SCBD score of the lost non-dbtaxa to the modern dbtaxa with a Wilcoxon statistic test. Phylogeny of the assigned taxa was retrieved using the 'V.Phylomaker.2' R package that implements mega-trees of vascular plants (GBOTB.extended.TPL, version 0.1.0[108]). Cophenetic distances between taxa were measured with the cophenetic.phylo() function from the ape R package (version 5.6-2[109]). The average distances between the lost non-dbtaxa, and the other taxa were compared with a Wilcoxon statistic test. To investigate vegetation composition changes, the dbtaxa and non dbtaxa were categorised within four functional plant types based on their best taxonomic assignment: grasses, rushes and grass-like sedges (Poaceae, Juncaceae and Cyperaceae) as graminoids; forbs, which are other herbs; shrubs; and trees (Supplementary Data 3). We compared the proportion of the lost non-dbtaxa relative to the total taxa per functional plant type. Significant differences between groups were tested by pairwise comparison with Bonferroni correction. The same methods were used to compare families. Based on the taxonomic name, taxa were grouped into different plant families. Only families with a minimum of five non-dbtaxa were investigated.

## Reporting summary

Further information on research design is available in the Nature Portfolio Reporting Summary linked to this article.

## Data availability

All data needed to evaluate the conclusions in the paper are present in the paper and/or the Supplementary Information. Final *sed*aDNA data used are given as Supplementary Data 3. All newly generated raw *sed*aDNA sequence data are deposited in the European Nucleotide Archive (ENA) (www.ebi.ac.uk/ena/browser/home) under the INSDC accession number: PRJEB76237. Input files needed to run the different scripts are given in Supplementary Data 5.

## Code availability

The bioinformatic scripts for the design and build of the SibAla_2023 database are available as a github project deposited under accession code https://doi.org/10.5281/zenodo.14033298. The OBITools3 pipeline and R scripts used for processing the data for the entire pipeline are available as a second github project deposited under accession code https://doi.org/10.5281/zenodo.14033305.

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

## Acknowledgements

We thank T. Böhmer and J. Klimke at the Alfred Wegener Institute Helmholtz Centre for Polar and Marine Research for technical support and Y. Lammers at The Arctic University Museum of Norway for critical reading of this manuscript. We thank C. Jenks at the University of Bergen for proofreading this manuscript. This project has received funding from the Priority Programme 'International Continental Drilling Program' of the German Research Foundation and the European Research Council (ERC) under the European Union's Horizon 2020 research and innovation programme (grant agreement no. 772852) and the Initiative and Networking Fund of the Helmholtz Association. Also received funding from the European Research Council (ERC) under the European Union's Horizon 2020 research and innovation programme (grant agreement No 819192). This project was also funded the Federal Ministry of Education and Research (BMBF SQUEEZE project). The sediment records from the lakes Levinson Lessing and Emanda were retrieved within the framework of the project PLOT (Paleolimnological Transect) which was funded by the German Federal Ministry for Education and Research (grant no. 03G0859). We acknowledge the support by the Open Access publication fund of Alfred Wegener Institute Helmholtz Centre for Polar and Marine Research.

## Author contributions

Conceptualisation: J.C., U.H. Methodology: U.H., J.C., K.R.S. Material acquisition: U.H., B.K., B.D., M.M., B.W., L.P., J.R., Y.H. Investigation: J.C., S.L., Y.L., U.H., K.R.S. Visualisation: J.C., S.L., U.H. Supervision: U.H. Writing—original draft: J.C., U.H., K.R.S., S.L., I.G.A. Writing—review & editing: J.C., K.R.S., S.L., Y.L., I.G.A., B.K., B.D., M.M., B.W., L.P., J.R., Y.H., U.H.

## Funding

## Competing interests

The authors declare no competing interests.
