## [Peer Review file · Nature Communications]

Potential plant extinctions with the loss of the Pleistocene mammoth steppe

Corresponding Author: Professor Ulrike Herzschuh

Version 0:

Reviewer comments:

Reviewer #1

(Remarks to the Author)

This paper uses a spatially expansive selection of sediment cores distributed across eastern Siberia and Alaska to assess potential vegetation extirpations or extinctions using sediment aDNA. By pooling multiple DNA datasets, the authors had improved taxon assignments, and identified ~60 potential sequences that had no modern match. Their aim was to understand the potential role of megafaunal extinctions and abrupt climate change on vegetation turnover and/or taxon extirpations or extinctions during a period of widespread global change.

While I find the study's premise exciting and the broader topic is of high relevance, this paper is undermined by several flaws in the framing and the approach. These could perhaps be ameliorated, but there will still be some limits that need better qualification and transparency. The paper could be better organized, and it often feels as though the megafauna story is tacked on as an afterthought. There are also a number of key references missing (even understanding the brevity necessary for a paper of this type). As a result, the overall scholarship fails to position the paper within broader dialogues about drivers of plant diversity, extinction, and megaherbivore ecology. This in turn undermines the arguments, as some statements (noted throughout below) are either incorrect or are more nuanced or complex than is being represented here (for example, our understanding of the underpinnings of extinction risk is much more nuanced and dynamic than is represented here).

As an example, the authors state that "we identified potential plant taxa loss events that happened in parallel to both the Plesistocene mammoth-steppe loss (vegetation turnover) and the megafauna extinction showing the importance of of past trophic interactions and potential co-extinction events in parallel to the loss of past biota," but this argument is not well-scaffolded throughout-- the setup and supporting literature are missing, and the discussion focuses more on the approach, at the expense of the ecological questions this paper is attempted to grapple with. This is a fixable problem, but it's going to take a lot of engagement with the literature and some reframing of the text.

Perhaps more critically, though, I have two major concerns about the experimental design. While I'm enthusiastic about these questions and the promise of the methods in addressing them, it just may not be possible to do what you're trying to do, and at a minimum, this text needs a lot of qualifying statements. My first concern is that the taphonomy of sedaDNA does not allow for us to say anything about extinction or even extirpation at this coarse spatial resolution. We can certainly say that some sequences were locally extirpated, but this requires a more up-front framing of what "local" means in a sedaDNA record (i.e., within the catchment area of the lake, within the proxy's representation, etc., which is not large for sedaDNA). In terms of spatial grain, eight lakes spanning a broad geographic range is not sufficient to capture biodiversity loss, especially when the source area of sedaDNA is so small compared with, e.g., pollen (which has too coarse a taxonomic resolution).

Secondly, I am not convinced that an incomplete ASV match means that the taxon has been lost, and while the authors do identify some caveats, many are missing. There could be some within-species variation in sequences due to local adaptation, and we know that many reference datasets have taxonomic errors (i.e., the assignments are not accurate because the species was incorrectly identified). But more importantly, there are gaps in our reference datasets, particularly within the region of this analysis. While it may be the case that the 60 taxa identified as "missing" have actually disappeared (though see my previous point about spatial grain), it's just as likely that you have an incomplete library problem. I may have missed it, but this appears not to have been discussed at all, even in the caveats section.

Here's why this gets tricky: the authors identify an extinction window of 19,000 to 9,000 BP, but the problem is, this

corresponds with the period of highest climate-driven taxonomic turnover due to climate-driven range shifts.

As far as calculating extinction rates, this is not even typically done for the entire Quaternary, let alone a particular time slice within it. The reason the denominator for extinction rates is a million years is because there is variability at shorter timescales. Elevated rates above background are just hard to quantify at timescales this small.

Ultimately, while I'm really excited by the questions this paper asks, and I am really supportive of the authors using some interesting and novel approaches to gain traction, I'm just not convinced that this study isn't just showing what we already know, but in a more taxonomically detailed way: that megafaunal losses are associated with the collapse of the mammoth steppe and widespread vegetation turnover. Without confidence that these are actual extirpations, rather than range shifts or contractions, it's not clear what we've gained by adding the higher taxonomic resolution of sedaDNA. I would suggest reframing this story as being about sequence losses, because loss of genetic diversity is, at a minimum, locally important and a precursor to extinction.

As a final note, using pollen-based temperature reconstructions to explain aDNA-inferred taxon losses is circular, because both proxies come from the same place (plants). While I deeply sympathize with the lack of regional resolution independent paleoclimate records, some do exist (including from some of the sites used in this study), so I'm curious why the authors chose not to include those records? Or why not use paleoclimate simulations? And rather than MAT, I would recommend using bioclimatic variables that have been shown to be the most important for limiting plant ranges, such as mean temperature of the coldest month and precipitation (both means and seasonality). This has the potential to be far more informative. While I deeply appreciate that such data are limited for the paleorecord, we can definitely do better than the circularity of pollen-derived MATs. At a minimum, I'd like to see some alternatives used here.

Some line-by-line comments follow:

31 While patterns of extinction remain poorly constrained (due in part because of poor biodiversity estimates) I don't think it's accurate to say that the drivers of extinction remain unknown. There's a broad literature about the causes of extinction that this statement glosses over.

35 This is a misrepresentation of the literature. The paper cited (4) does not argue that estimates are unreliable, but rather points out a discrepancy between predictions and reality. There are a number of potential mechanisms beyond data quality that are explored in the paper and elsewhere, and this statement and its citations aren't doing that literature justice.

36 This statement is a bit misleading, and is missing a number of key references on the impacts of climate change on extinction. The paper cited doesn't actually refer to debates, but rather points out that climate change is seldom included in the assessment of extinction risk in IUCN Red list criteria. There have been numerous studies, both for individual species and broadly across clades, that seek to forecast extinction risk due to climate change. Those paper should be cited at a minimum.

38 This would be a great place to cite the conservation paleobiology literature.

46 But see also: Winter et al., 2009, PNAS (<https://doi.org/10.1073/pnas.0907088106>), Purvis 2008, Annual Rev. of Ecology and Systematics (<https://www.annualreviews.org/doi/abs/10.1146/annurev-ecolsys-063008-102010>), Davies et al., 2011 (<https://journals.plos.org/plosbiology/article?id=10.1371/journal.pbio.1000620>)...and others which contradict this statement.

54 The question of whether there were plant extinctions driven by megafaunal extinctions is intriguing, but needs more development here. Most ecosystems that lost megaherbivores did retain some examples, and we don't see the disappearance of biomes per se.

56 I don't typically see a hyphen between "mammoth" and "steppe" and would suggest removing it.

57 This is unclear. If the species range shifts are plant species range shifts, that's a mechanism to avoid extinction during climate change, no?

58 This paper explores forecasted extinction risks, but there are a number of studies on the impacts of Quaternary climates and megafaunal extinctions on plant ranges and extinctions, and the framing in the next sentence "this tremendous climate and ecosystem change" implies that this reference [20] is relevant to that.

60 I would cite Botkin et al., 2007, BioScience (<https://doi.org/10.1641/B570306>), which refers to a "Quaternary conundrum." There is also deeper-time literature that discusses the resilience of plants in the extinction record that would be worth citing in this intro, too, such as McElwain & Punyasena 2007, TREE (<https://doi.org/10.1016/j.tree.2007.09.003>) and Scott Wing's chapter on plants and mass extinctions in *Extinctions in the History of Life*. I would also make it clear that it's not an absence of study driving the few documented plant extinctions at the Pleistocene-Holocene transition. This has been a widely recognized phenomenon. It's not that people haven't looked.

65 Do we have the ability to say that this transition resulted in a loss of diversity per se? This is a great place to point out the taxonomic coarseness of pollen data, and how aDNA can improve diversity estimates.

69 *P. critchfieldii* wasn't necessarily common – the plant macrofossils were found in the southeastern United States, and it

was likely already in low abundance based on the available evidence (though this could reflect a paucity of sites).

70 Large physical size or range size? Both?

72 This is confusing in the context of the paleorecord. When talking about pollen, plant macrofossils, and DNA, size has little bearing on proxy-based reconstructions of species?

72 This has been extensively researched in modern systems, though. It's also unclear what this statement means in the context of this paragraph – if you're referring to paleo extinctions, those apparently have not occurred, so this isn't a knowledge gap. If you're referring to modern extinctions, that's a subject of active research, and this point also doesn't follow from the preceding sentences. It feels like this section contains several different ideas and it's unclear how they are connected, especially as you weave in modern and paleo phenomena and data sets.

77 There have been several studies recently that use aDNA to assess plant diversity changes within the mammoth steppe at the local scale, which is "sub-continental," so I would clarify the spatial extent you're referring to here.

79 It appears that the majority of the pollen types in this dataset are at the genus level, not family level?

90-94 It's not entirely clear what's meant by this paragraph – do you mean that aDNA sequences can be used as a proxy to reconstruct community and range dynamics? That seems straightforward, but this should be clarified, and also include some discussion about how relative abundance cannot (yet, at least) be inferred from sedaDNA records.

94 How can you distinguish between plants that are missing in reference libraries and plants that have gone extinct prior to the establishment of those libraries? Are there particular bioinformatic approaches?

104 No hyphen needed between "lake" and "sediment."

128 I would like to see the actual proportions listed across these time bins (i.e., quantify the increase).

140 I interpret "modern time slice" to mean sediments from core tops, but if that's not the case, this needs to be more clear. Readers will not have read the methods at this point in the paper, so clarity here is especially important.

Fig. 2 It seems weird to include carnivores (Homotherium, Panthera) in this figure. I would explain the justification or focus just on herbivores.

289 Possibly, though there are plenty of counter-examples of rare taxa that have survived as rare taxa for a very long time, and many abundant plant taxa went extinct in Europe (see: Svenning and Skov's work on European tree extinctions).

293 The use of present tense here and in preceding sentences muddies the waters a bit. I would suggest past tense for results, which will make it more clear that you are talking about your results and not generalisms.

303 This is potentially significant, and so I would like to see some discussion about why that is. Is it a rarefaction artifact (i.e., there were more forb and graminoids, so they are statistically likelier to go extinct)?

315 This is a crucial point that should be highlighted sooner in the text, but it also needs some qualification, because the databases themselves are incomplete, which is a point that is glossed over in this text.

342 Or from Asparagaceae that just haven't been added yet.

357 This is another oversimplification that needs some unpacking or qualification. The underpinnings of ecological stability (or resistance and resilience) have been extensively studied (though questions still remain), and diverse ecosystems are often more stable even in the face of abiotic instability. And on longer timescales, environmental variability can be a speciation pump that drives diversification. We don't even know whether the tropics have more diversity because they are stable (museum) or because they drive diversification (cradle).

359 "It" has an unclear antecedent here.

362 Be careful about words like dramatic, which are normative. Also, how does this interpretation stand up to the fact that there have been multiple glacial cycles in the Quaternary? Shouldn't these be taxa that have already adapted to some degree to high amounts of environmental variability at high latitudes (both seasonally and on glacial-interglacial cycles)?

365-368 This is bordering on tautology – there was more turnover because there was more turnover.

369-371 This final sentence feels vague and underdeveloped – it's not entirely clear what the conclusion is meant to be.

375 Should be "Bering land bridge." How do you differentiate between Beringia, Siberia, and Alaska (if Beringia is inclusive of both)?

387 Some of these cores have been published previously – this statement implies that these are new cores. Is that the case,

or was previously existing sediment used? If the latter, this needs qualifying.

Reviewer #2

(Remarks to the Author)

In their paper 'Potential plant extinctions with the loss of the Pleistocene mammoth-steppe' Courtin et al. use metabarcoding to investigate loss of plant species in arctic Russia and Alaska during the transition from the Pleistocene to the Holocene.

They do so by characterizing ASVs according to their similarity to their reference database, and, in cases where ASVs appears distinct from both modern time slices and the reference database, they characterize that ASV as potentially extinct. I like the overall angle of the manuscript, and I agree with the authors that plant extinctions are terribly understudied.

However, my main concern with this manuscript also relates to the focus on extinctions in the paper. I am not convinced that what the author quantify is actually extinctions. Could it not simply be range shifts, in combination with incomplete databases? I think the paper could benefit from a slight rewriting, with less emphasis on extinctions and more focus on community changes and perhaps loss of diversity.

Furthermore, it might be beyond the scope of this paper, but the conclusions here could be strengthened by including more data. I know that the paper 'Fifty thousand years of Arctic vegetation and megafaunal diet' has a large trnL-gb dataset from the same area. Perhaps this dataset could be merged with the dataset at hand?

That being said, I think that this is an important paper with a novel approach to analyse these increasingly large and complicated metabarcoding datasets, and I think that it is a good fit for Nature communications.

Minor comments:

Fig 1:

I think that the map from figure 4 should be moved to this figure.

Furthermore, I would recommend splitting figure 1c into two panels, one for candidate taxa and one for dbtaxa. This would make it easier for the reader to appreciate subtle differences between the two. It is also currently a bit confusing that the same colours are repeated twice.

Line 140: This assumption is almost certainly incorrect. If you sampled more in the modern time slice, you might find many of these 'extirpated' taxa.

Line 146: 'the proportion of candidate taxa present in a time-slice decreases through time' I don't think that this is supported by the data in Fig 1c.

Fig 2:

It is not clear how the confidence threshold of 80% was chosen.

Furthermore, it would be interesting to also have the Greenland ice core data to compare with (perhaps overlaid on fig 2d).

Fig 4:

This figure gives a good overview of the analytical approach. Perhaps it should be moved, so that this becomes the new figure 1.

Reviewer #3

(Remarks to the Author)

The authors have provided an important new contribution to our understanding of plant extinction and extirpation in the former "Mammoth Steppe" landscapes of the northern high latitudes. The geographic range of their study makes the work particularly valuable for scholars focusing on the Late Quaternary Extinctions and the patterns and processes driving that episode in Earth history. The dynamics of plant communities during and following the LQE are still not well understood. The authors have added new depth to this particular field by taking an expansive view across multiple Siberian/Alaskan biomes as they exist today. Their analysis of sedimentary DNA taken from lake cores drawn from multiple sites reveals significant patterns of plant losses correlating with known waves of large herbivore extinctions leading to the disappearance of the Mammoth Steppe. They find that rates of plant losses may have reached 120 times the background rates of extinction. Their results show these episodes correlate with rapid environmental shifts of the Late Pleistocene. Such instability becomes most pronounced in the high latitudes. From this, the authors make a good case for the relevance of their analysis of paleoecological proxy data, given that climate and environmental change in these regions is proceeding considerably more rapidly today. This report is building on a growing body of work in this area that is well-referenced here.

My comments below are minor but mainly focus on the clarity of the writing so as to make it more accessible to a wider body of readers.

From the Abstract and in the Introduction and other sections, plant losses are characterized as "extirpations", and "potential extinctions", but these terms should be clearly defined and explained as they apply to the findings presented here. I am sure the authors have precise meanings for each of these terms, and I assume neither means actual extinction. It would be most helpful to the reader if these terms were explained early on in the Introduction. For the Abstract, I would suggest wording that briefly draws out the meaning of each, rather than introducing both terms parenthetically (line 19). Since the main thrust of

the subsequent data interpretation hinges on these two concepts, it is important to have them made clear at the outset. Beta diversity is another term that although used in certain areas of community ecology, is not commonly used in paleoecological studies. It would be helpful to the reader if the authors explained this concept early on in the Introduction, as it also is pivotal to their data interpretation.

In the Discussion section, (line 262 and following) in referring to the two waves of megafaunal extinction, the first during the Late Glacial Maximum; and the second peaking between 15,000 and 10,000 years bp., we learn that “estimated plant extirpation” peaking around 13,000 bp, (in the middle of this time interval) does not correlate with megafaunal losses. Nevertheless, “potential plant extinction” does correlate with this second wave. At first, this seems paradoxical. No doubt this is an instance of the particular uses of the terms as outlined above in 1).

It would be useful to have at least a brief description of the geographic setting of each of the sites where the cores were taken. As it is, the map in Figure 4 and Table 1 appear to be the only places where the reader can find this information, yet this would involve following each of the references listed in the far right column of Table 1.

Version 1:

Reviewer comments:

Reviewer #2

(Remarks to the Author)

In their resubmitted article Courtin et al have used the GBIF database to estimate the expected number of absent species in their SibAla_2023 database. Using simulations based on this fraction the authors are able to estimate how large a proportion of the 'lost' taxa that are probably due to incomplete databases. Based on this, the authors estimate that 17 out of 60 species are potentially extinct. I like their approach of using simulations to account for the incomplete reference libraries, and, as such my main concern about the paper have been addressed.

I still have a couple of comments:

Maybe I have misunderstood how this simulation works, but the simulation suggests that 42/60 species are due to incomplete databases. What is the error margin on this estimate? Because if the error margin is, say +/- 20, there is no evidence for extinctions at all.

Given that the main conclusion of the paper has been changed dramatically (i.e. from “extinction rate was up to 120X above bg” to “extinction rate were below modern estimates”), I think the title should be updated to reflect this. Perhaps the title could start with “Low extinction rates during the loss [...]” or “Plant resilience during the [...]”

And speaking of the significant change in conclusion, I would like to flag that I am still not 100% convinced of the results. It is a very difficult task the authors have set themselves, estimating species loss. Having worked with metabarcoding data myself, I have sometimes found that seemingly small changes in the filtering stage (for example changing the cutoff from 100 read to 10 reads) can have dramatic effects on downstream analysis. But I guess only time will tell if these conclusions stand in the future.

Minor comments:

My last comment on what is now Supplementary fig 3 still stands. To me it does not make sense to show the non-db and db data stacked in this way, as it veils all patterns for the top category ('db taxa' in this case).

Kind regards,
Frederik Seersholm, University of Copenhagen

(Remarks on code availability)

Code was not available for review as the github depositories are not publicly available (yet). It should be noted that according to the authors "they can be provided upon request" during the review process.

Reviewer #3

(Remarks to the Author)

The authors have addressed each one of my comments and concerns. They have reworked the text accordingly.

Reviewer 1's comments are incisive and thorough, highlighting some fundamental weaknesses in the structure of the author's overall argument and how the results have been brought to bear in support of that argument. Reviewer 1 nevertheless expresses enthusiasm for the ambitious aims and innovative approach to a complex problem. In this, I agree; I find the revised manuscript better positioned to begin tackling that problem.

Regarding the major comments, I find the authors have addressed each one appropriately, restructuring their argument and engaging a broader body of the literature. Importantly they have incorporated the called for caveats and qualifications. The limitations of using existing reference databases and their way of addressing this are discussed in the paragraph following

line 350.

The problem of the plant extinction window of 19,000 to 9,000 BP coinciding with the highest rate of climate-driven range shifts is addressed in 511ff where the authors identify a significant time lag before potential extinctions happen.

Noting the comment on the circularity of pollen-based temperature reconstructions to explain taxon losses derived from ancient DNA, I find the authors have made a good case for retaining this temperature reconstruction. While it is true that both proxies derive from plants, it is not the same array of plants. As observed by Jorgensen, et al. 2012, pollen, macrofossil, and sedaDNA data are complementary rather than overlapping.

Among the line-by-line remarks, I agree that mammoth steppe should not be hyphenated. Dale Guthrie coined the term and did not use a hyphen (although the title of Murchie, et al. 2021 uses a hyphen in Nature Communications).

I find the revised manuscript to be suitable for Nature Communications.

(Remarks on code availability)

Potential plant extinctions with the loss of the Pleistocene mammoth steppe

Answer to the reviewers

Jérémy Courtin¹, Kathleen R. Stoof-Leichsenring¹, Simeon Lisovski¹, Ying Liu¹, Inger Greve Alsos², Boris K. Biskaborn¹, Bernhard Diekmann¹, Martin Melles³, Bernd Wagner³, Luidmila Pestryakova⁴, James Russell⁵, Yongsong Huang⁵ & Ulrike Herzschuh^{1,6,7*}

¹Polar Terrestrial Environmental Systems, Alfred Wegener Institute Helmholtz Centre for Polar and Marine Research, Germany

²The Arctic University Museum of Norway, UiT - The Arctic University of Norway, Norway

³Institute of Geology and Mineralogy, University of Cologne, Germany

⁴Institute of Natural Sciences, North-Eastern Federal University of Yakutsk, Russia

⁵Department of Earth, Environmental and Planetary Sciences, Brown University, USA

⁶Institute of Environmental Science and Geography, University of Potsdam, Germany

⁷Institute of Biology and Biochemistry, University of Potsdam, Germany

*Correspondence

Ulrike Herzschuh, ulrike.herzschuh@awi.de

Point-by-Point response Reviewer1:

Reviewer1 major comment: This paper uses a spatially expansive selection of sediment cores distributed across eastern Siberia and Alaska to assess potential vegetation extirpations or extinctions using sediment aDNA. By pooling multiple DNA datasets, the authors had improved taxon assignments, and identified ~60 potential sequences that had no modern match. Their aim was to understand the potential role of megafaunal extinctions and abrupt climate change on vegetation turnover and/or taxon extirpations or extinctions during a period of widespread global change.

While I find the study's premise exciting and the broader topic is of high relevance, this paper is undermined by several flaws in the framing and the approach. These could perhaps be ameliorated, but there will still be some limits that need better qualification and transparency. The paper could be better organized, and it often feels as though the megafauna story is tacked on as an afterthought. There are also a number of key references missing (even understanding the brevity necessary for a paper of this type). As a result, the overall scholarship fails to position the paper within broader dialogues about drivers of plant diversity, extinction, and megaherbivore ecology. This in turn undermines the arguments, as some statements (noted throughout below) are either incorrect or are more nuanced or complex than is being represented here (for example, our understanding of the underpinnings of extinction risk is much more nuanced and dynamic than is represented here).

As an example, the authors state that “we identified potential plant taxa loss events that happened in parallel to both the Plesistocene mammoth-steppe loss (vegetation turnover) and the megafauna extinction showing the importance of of past trophic interactions and potential co-extinction events in parallel to the loss of past biota,” but this argument is not well-scaffolded throughout-- the setup and supporting literature are missing, and the discussion focuses more on the approach, at the expense of the ecological questions this paper is attempted to grapple with. This is a fixable problem, but it's going to take a lot of engagement with the literature and some reframing of the text.

Our response: Thank you for your comment. We believe the strength of this manuscript lies in its novel systematic approach to handling *sedaDNA* data. Following your advice, we have added additional citations to enhance the ecological discussion. The first paragraphs of the discussion now emphasize the robustness of our results while addressing caveats and potential limitations. Additionally, we have revised the rest of the discussion to focus more on the ecological implications rather than the methodological approach.

New text: line 305 to 385 for the discussion of limitations and robustness.

Reviewer1 major comment: Perhaps more critically, though, I have two major concerns about the experimental design. While I'm enthusiastic about these questions and the promise of the methods in addressing them, it just may not be possible to do what you're trying to do, and at a minimum, this text needs a lot of qualifying statements. My first concern is that the taphonomy of *sedaDNA* does not allow for us to say anything about extinction or even extirpation at this coarse spatial resolution. We can certainly say that some sequences were

locally extirpated, but this requires a more up-front framing of what "local" means in a *sedaDNA* record (i.e., within the catchment area of the lake, within the proxy's representation, etc., which is not large for *sedaDNA*). In terms of spatial grain, eight lakes spanning a broad geographic range is not sufficient to capture biodiversity loss, especially when the source area of *sedaDNA* is so small compared with, e.g., pollen (which has too coarse a taxonomic resolution).

Our response: Thank you for your comment. We conducted new tests and made additional efforts to highlight the robustness of our approach in demonstrating past plant extinctions. We now provide a minimum and maximum extinction rate while remaining conservative in our estimates. The main text now includes results showing that the lost non-dbtaxa are not only a fraction missing from the SibAla_2023 reference database. Furthermore, we tested whether the small fraction of lost non-dbtaxa matching to other databases correspond to related species that might have survived in refugia. Additionally, we have added new paragraphs in the discussion that extensively presents the potential limitations of our approach and the taphonomy of *sedaDNA*. We used all available cores that covered the Pleistocene to Holocene transition with good sample resolution, ensuring samples from each 2000-yr time slice over the last 28,000 years. While more sites should be investigated to allow a more robust estimate, our study of seven sites does confirm the general assumption that Pleistocene plant extinction rates are far lower than those for mammals. We hope that our study can be used as a baseline with an established methods on how extinction events can be explored using *sedaDNA* datasets for future studies that will incorporate more sites and better reference databases and encourage the community to work in that direction.

New text: line 182 to 224 for the description of further analysis to improve the robustness of our approach and the new estimated range of potentially extinct fraction.

Reviewer1 major comment: Secondly, I am not convinced that an incomplete ASV match means that the taxon has been lost, and while the authors do identify some caveats, many are missing. There could be some within-species variation in sequences due to local adaptation, and we know that many reference datasets have taxonomic errors (i.e., the assignments are not accurate because the species was incorrectly identified). But more importantly, there are gaps in our reference datasets, particularly within the region of this analysis. While it may be the case that the 60 taxa identified as "missing" have actually disappeared (though see my previous point about spatial grain), it's just as likely that you have an incomplete library problem. I may have missed it, but this appears not to have been discussed at all, even in the caveats section.

Our response: Thank you for your comment. Taking this into account, we performed extensive tests to highlight the robustness of our results, such as checking the database coverage and the hypothesis of refugia. We now provide a range of potential extinction from 1.7 to 5.9 E/MSY which are above background extinction but below modern estimates. Additionally, we restructured all sections of the manuscript to better emphasize the strengths and caveats of our study. In the introduction, we provide further information on the limitations of existing methods and how our approach addresses these issues. In the discussion, the first paragraphs now focus

on highlighting our main results and placing them in the context of known biases and caveats without diminishing their significance.

New text: line 87 to 11 for the introduction of limits of current methods. Line 305 to 385 for the discussion of limitations and robustness of our approach.

Reviewer1 major comment: Here's why this gets tricky: the authors identify an extinction window of 19,000 to 9,000 BP, but the problem is, this corresponds with the period of highest climate-driven taxonomic turnover due to climate-driven range shifts.

Our response: Thank you for your comment. This is true, and indeed it may be hard to distinguish turnover from extinction, as both are likely to take place at the same time. If all loss of non-dbtaxa was solely due to range shift and thus represent extirpation, we would expect the proportion of loss to be equal for dbtaxa and non-dbtaxa. The somewhat higher loss of non-dbtaxa may be ascribed that they in general are more rare and thus difficult to detect. Alternatively, they may represent global extinction, as suggested by the fact that most are missing from modern global databases. We have now made this point clearer in the discussion. Further, species turnover implies species replacement and loss in the study area that can be due to climate change and range shifts. With ongoing climate-driven changes, we observe that plant species are both changing range and going extinct during modern times. Therefore, we hypothesize that a fraction of past plant species facing similar stresses and constraints could also have gone extinct in parallel to others shifting range. Additionally, we have included another test to highlight the robustness of our approach and for example while checking among the lost non-dbtaxa, we estimate that only a small fraction of these species matches other databases and could have survived in refugia (6 taxa) and therefore lower our estimate for potential plant extinction now ranging from 17 to 59 taxa.

New text: Line 333 to 340 for the clarification in the discussion. Line 211 to 221 for the potential refugia check.

Reviewer1 major comment: As far as calculating extinction rates, this is not even typically done for the entire Quaternary, let alone a particular time slice within it. The reason the denominator for extinction rates is a million years is because there is variability at shorter timescales. Elevated rates above background are just hard to quantify at timescales this small.

Our response: Thank you for your comment. The extinction rates we calculated are also used on ongoing extinctions as highlighted in Cowie et al., 2022. In this case, the rates are derived from short-term observations (taxonomic identification has only existed for a few centuries), and not all taxa have been described, so there is only an estimate of the total number of taxa. We assume that both modern data and past *sed*aDNA-based data face similar limitations. Therefore, we used these rates to provide a basis for comparison between extinction rates that occurred during the transition between the Pleistocene and the Holocene and modern rates.

Reviewer1 major comment: Ultimately, while I'm really excited by the questions this paper asks, and I am really supportive of the authors using some interesting and novel approaches to gain traction, I'm just not convinced that this study isn't just showing what we already know,

but in a more taxonomically detailed way: that megafaunal losses are associated with the collapse of the mammoth steppe and widespread vegetation turnover. Without confidence that these are actual extirpations, rather than range shifts or contractions, it's not clear what we've gained by adding the higher taxonomic resolution of sedaDNA. I would suggest reframing this story as being about sequence losses, because loss of genetic diversity is, at a minimum, locally important and a precursor to extinction.

Our response: Thank you for your comment and interest in our study. The method we applied was specifically designed to identify patterns of appearance and disappearance of plant taxa over time, using a community approach rather than relying solely on the best assigned taxonomic name. We decided to focus on estimating potential extinction rates and conducted further tests to underscore the robustness of our results. We hope these additions convince you that we could identify potential plant extinctions during this period, as indicated by the loss of DNA sequence signals absent from modern DNA databases. Additionally, the marker used in our study is effective for identifying plant taxa using small (ancient) DNA fragments but is limited when investigating population genetics. For such investigations, other approaches, such as metagenomics with hybridization capture or shotgun sequencing, would be recommended. We understand and agree that the framing of the study needed improvement. To address this, we have completely restructured the introduction, adding more literature to emphasize previous work in the field of paleoecology and highlighting nuances and unresolved questions. Additionally, we have worked to provide more ecological context to the study, thereby enhancing its framework. Similar restructuring has been performed in the discussion section to better scaffold our arguments. This section has been revised with new paragraphs to highlight some limitations while still pinpointing the main results this study can offer within the constraints of sedaDNA datasets.

New text: line 182 to 224 for the new results presented to improve the robustness of our results and refine the potentially extinct taxa rates to a range. Line 58 to 62, 71 to 73 and 76 to 81 for added literature in the introduction.

Reviewer1 major comment: As a final note, using pollen-based temperature reconstructions to explain aDNA-inferred taxon losses is circular, because both proxies come from the same place (plants). While I deeply sympathize with the lack of regional resolution independent paleoclimate records, some do exist (including from some of the sites used in this study), so I'm curious why the authors chose not to include those records? Or why not use paleoclimate simulations? And rather than MAT, I would recommend using bioclimatic variables that have been shown to be the most important for limiting plant ranges, such as mean temperature of the coldest month and precipitation (both means and seasonality). This has the potential to be far more informative. While I deeply appreciate that such data are limited for the paleorecord, we can definitely do better than the circularity of pollen-derived MATs. At a minimum, I'd like to see some alternatives used here.

Our response: Thank you for your comment. Regarding the use of pollen-based temperature reconstruction, we tested several proxies for past climate changes, including simulated temperature and precipitation estimates from records covering the study area (Dallmeyer et al.,

2022, 10.1038/s41467-022-33646-6) and temperature and precipitation estimates from pollen-based reconstruction (both MAT and WA-PLS, Herzsuh et al. 2022, 10.5194/essd-14-3213-2022). After concluding that both simulated and pollen-based reconstructions yielded similar results (i.e., correlation to plant taxa loss), we chose to use the pollen-based data. This decision was based on known evidence that pollen-based reconstructions generally align better with expected regional changes than simulated data. This is further detailed in the supplementary note 6.

Some line-by-line comments follow:

Reviewer1 minor comment: 31 While patterns of extinction remain poorly constrained (due in part because of poor biodiversity estimates) I don't think it's accurate to say that the drivers of extinction remain unknown. There's a broad literature about the causes of extinction that this statement glosses over.

Our response: Thank you for your comment. We agree that our point was not clear and may have been misleading. To address this, we have completely rewritten the introduction to improve the overall framework of the study and clarify our approach.

Reviewer1 minor comment: 35 This is a misrepresentation of the literature. The paper cited (4) does not argue that estimates are unreliable, but rather points out a discrepancy between predictions and reality. There are a number of potential mechanisms beyond data quality that are explored in the paper and elsewhere, and this statement and its citations aren't doing that literature justice.

Our response: Thank you for your comment. As previously mentioned, we agree that the first version of the introduction did not adequately highlight previous work in the field. We have completely reworked and restructured the introduction to address this issue.

Reviewer1 minor comment: 36 This statement is a bit misleading, and is missing a number of key references on the impacts of climate change on extinction. The paper cited doesn't actually refer to debates, but rather points out that climate change is seldom included in the assessment of extinction risk in IUCN Red list criteria. There have been numerous studies, both for individual species and broadly across clades, that seek to forecast extinction risk due to climate change. Those paper should be cited at a minimum.

Our response: Thank you for your comment. We believe that we have addressed this concern and clarified our introduction in its revised version by adding further literature.

Reviewer1 minor comment: 38 This would be a great place to cite the conservation paleobiology literature.

Reviewer1 minor comment: 46 But see also: Winter et al., 2009, PNAS (<https://doi.org/10.1073/pnas.0907088106>), Purvis 2008, Annual Rev. of Ecology and Systematics (<https://www.annualreviews.org/doi/abs/10.1146/annurev-ecolsys-063008-102010>), Davies et al., 2011

(<https://journals.plos.org/plosbiology/article?id=10.1371/journal.pbio.1000620>)...and others which contradict this statement.

Our response: Thank you. With your general comments in mind, we now implemented further citations to better highlight previous conservation paleobiology literature and nuanced our original introduction.

New text: Added literature as recommended line 58 to 62 and 71 to 81 for example.

Reviewer1 minor comment: 54 The question of whether there were plant extinctions driven by megafaunal extinctions is intriguing, but needs more development here. Most ecosystems that lost megaherbivores did retain some examples, and we don't see the disappearance of biomes per se.

Our response: Thank you for your comment. Considering that the mammoth steppe biome lacks a modern analogue, except for potential relicts in the Altai mountain range, and given the simultaneous occurrence of megafaunal extinction during the transition between the Pleistocene and the Holocene, we believe this context scaffolds well the hypothesis of potential plant extinction.

New text: line 48 to 66.

Reviewer1 minor comment: 56 I don't typically see a hyphen between "mammoth" and "steppe" and would suggest removing it.

Our response: Thank you for your comment. We removed the hyphen throughout the manuscript.

Reviewer1 minor comment: 57 This is unclear. If the species range shifts are plant species range shifts, that's a mechanism to avoid extinction during climate change, no?

Our response: Thank you for your comment. It is well established that species possess specific geographic ranges, and they may attempt to track environmental changes if their niche changes with climate change. However, certain species may be unable to shift their ranges at a pace commensurate with the rapidity of environmental shifts. Consequently, they face the risk of range contraction, extinction, or complete habitat loss (Parmesan et al., 2005, 10.1175/1520-0477(2000)081 ; Urban, 2015, 10.1126/science.aaa4984).

Reviewer1 minor comment: 58 This paper explores forecasted extinction risks, but there are a number of studies on the impacts of Quaternary climates and megafaunal extinctions on plant ranges and extinctions, and the framing in the next sentence "this tremendous climate and ecosystem change" implies that this reference [20] is relevant to that.

Our response: Thank you for your comment. We reworked the introduction and therefore this sentence do not exist anymore. This reference is now cited as 13.

New text: line 41 to 42 for new citation of this reference.

Reviewer1 minor comment: 60 I would cite Botkin et al., 2007, BioScience (<https://doi.org/10.1641/B570306>), which refers to a “Quaternary conundrum.” There is also deeper-time literature that discusses the resilience of plants in the extinction record that would be worth citing in this intro, too, such as McElwain & Punyasena 2007, TREE (<https://doi.org/10.1016/j.tree.2007.09.003>) and Scott Wing’s chapter on plants and mass extinctions in Extinctions in the History of Life. I would also make it clear that it's not an absence of study driving the few documented plant extinctions at the Pleistocene-Holocene transition. This has been a widely recognized phenomenon. It’s not that people haven’t looked.

Our response: Thank you for your comment. We acknowledge that previous studies were not adequately highlighted in the initial version of the manuscript. To address this, we have enhanced the introduction by conducting additional literature review and including an expanded bibliography including some papers you recommended.

New text: line 58 to 62, 76 to 78 and 81 to 84 are examples where we added the recommended literature.

Reviewer1 minor comment: 65 Do we have the ability to say that this transition resulted in a loss of diversity per se? This is a great place to point out the taxonomic coarseness of pollen data, and how aDNA can improve diversity estimates.

Our response: Thank you for your comment. We refer to two studies conducted in areas that exhibited a decline in plant richness during the transition between the Pleistocene and the Holocene (Courtin et al., 2021, 10.3389/fevo.2021.625096; Huang et al., 2021,10.3389/fevo.2021.763747).

New text: line 55 to 57.

Reviewer1 minor comment: 69 *P. critchfieldii* wasn’t necessarily common – the plant macrofossils were found in the southeastern United States, and it was likely already in low abundance based on the available evidence (though this could reflect a paucity of sites).

Our response: Thank you for your comment. We agree with your observation and have made the necessary modifications. Nonetheless, the confirmed extinction of only one plant species in the study area at the transition is of significant interest to us. This finding highlights the occurrence of plant extinction, a factor often underestimated in other studies, primarily due to the inadequate preservation of plant fossils that would enable species-level identification and potentially reveal extinct plant species. We believe that using this example serves as a solid foundation for formulating hypotheses regarding the potential extinction of other plant species during the critical transition from the last glacial period to the Holocene.

New text: line 58 to 60.

Reviewer1 minor comment: 70 Large physical size or range size? Both?

Our response: Thank you for your comment. We have rewritten this entire section, paying closer attention to clearly specify whether we are referring to large mammals or expansive geographic regions.

Reviewer1 minor comment: 72 This is confusing in the context of the paleorecord. When talking about pollen, plant macrofossils, and DNA, size has little bearing on proxy-based reconstructions of species?

Our response: Thank you for your comment. Once again, we have completely revised this section to enhance clarity. Your feedback has been invaluable in helping us recognize the need for this rewrite.

Reviewer1 minor comment: 72 This has been extensively researched in modern systems, though. It's also unclear what this statement means in the context of this paragraph – if you're referring to paleo extinctions, those apparently have not occurred, so this isn't a knowledge gap. If you're referring to modern extinctions, that's a subject of active research, and this point also doesn't follow from the preceding sentences. It feels like this section contains several different ideas and it's unclear how they are connected, especially as you weave in modern and paleo phenomena and data sets.

Our response: Thank you for your comment. You are entirely correct. It was never our intention to disregard previous work; rather, we aimed to emphasize the use of *sedaDNA* in investigating paleoextinction. In revising the introduction, we have tried to better highlight previous research on ongoing extinction. While considerable research has been conducted on past plant extinctions during mass extinction events, the extinctions that occurred at the transition between the last Glacial and Holocene did not reach such magnitude, whether in the mammoth steppe or other continents, especially for well-studied groups like megafauna. Similarly, the extent of potential plant extinction identified in our study does not approach the scale of mass extinction events. Therefore, we deemed it unnecessary to extensively introduce past research on mass extinctions.

Reviewer1 minor comment: 77 There have been several studies recently that use aDNA to assess plant diversity changes within the mammoth steppe at the local scale, which is “sub-continental,” so I would clarify the spatial extent you're referring to here.

Our response: Thank you for your comment. You are correct. We have now emphasized that this study utilizes multiple sites, providing a sub-continental scale perspective. Unlike other studies with multiple sites, our approach ensures sample resolution across all time series for all sites. For example, Wang et al. 2021 (10.1038/s41586-021-04016-x) lacked the necessary resolution to investigate taxa appearance and disappearance through time due to missing samples from recent times. This limitation prevented us from applying our method to their dataset, as we were unable to clearly identify Amplicon Sequence Variants and taxa absent from the modern time.

New text: line 87 to 89.

Reviewer1 minor comment: 79 It appears that the majority of the pollen types in this dataset are at the genus level, not family level?

Our response: Thank you for your comment. This has been modified and clarified.

New text: line 89 to 90.

Reviewer1 minor comment: 90-94 It's not entirely clear what's meant by this paragraph – do you mean that aDNA sequences can be used as a proxy to reconstruct community and range dynamics? That seems straightforward, but this should be clarified, and also include some discussion about how relative abundance cannot (yet, at least) be inferred from sedaDNA records.

Our response: Thank you for your comment. As previously mentioned, the introduction has been completely reworked in an effort to reframe the study and clarify our approach.

New text: line 109 to 111

Reviewer1 minor comment: 94 How can you distinguish between plants that are missing in reference libraries and plants that have gone extinct prior to the establishment of those libraries? Are there particular bioinformatic approaches?

Our response: Thank you for your comment. We realized that this aspect was not clear in the first version of the paper. Therefore, we have made a concerted effort to enhance the explanation of our new systematic method for identifying unknown taxa, particularly those absent from reference libraries, and detecting potential extinction signals. As detailed in the main text, this involves taxonomic identification using similarities ranging from 90% to 100%, which permits mismatches with the reference libraries. Subsequently, to mitigate potential PCR or sequence errors, we have introduced a novel co-occurrence-based approach and community detection method to filter out Amplicon Sequence Variants (ASVs) assigned with 90-99% confidence (non-dbASVs) that exhibit similar patterns to 100% assigned ASVs (dbASVs) and share comparable taxonomic names. While this approach is highly conservative, it enables us to retain only those candidate taxonomic names with distinctive co-occurrence patterns.

New text: line 596 to 609 and 619 to 629.

Reviewer1 minor comment: 104 No hyphen needed between “lake” and “sediment.”

Our response: Thank you for your comment. We removed the hyphen.

Reviewer1 minor comment: 128 I would like to see the actual proportions listed across these time bins (i.e., quantify the increase).

Our response: This section is not present anymore in the main text. You can find some extra information about the compositional changes in the supplementary note 4.3: In the samples from the Pleistocene (~27,000 to ~17,000 cal. yrs BP), 78.3% of the taxa were either forbs or graminoids, with more than 42% of the overall reads assigned. After the LGM, between ~15,000 and ~9,000 cal. yrs BP, there was a shift towards more shrub and tree taxa in the study

area. Between ~7,000 and ~1,000 cal. yrs BP, 29.5% of all taxa are assigned to shrubs or trees with more than 91% of the overall reads assigned. For the dbtaxa we see a small decrease in forbs (73.5% of all dbtaxa before 15,000 cal. yrs BP and 71% after 9,000 cal. yrs BP) and graminoids (11.6% to 8.1%) between the Pleistocene and the Holocene. The non-dbtaxa are generally more impacted with a decrease from 5.5% to 3.1% for graminoids and especially from 61.4% to 42.4% for forbs between before 15,000 cal. yrs BP and after 9,000 cal. yrs BP.

Reviewer1 minor comment: 140 I interpret “modern time slice” to mean sediments from core tops, but if that’s not the case, this needs to be more clear. Readers will not have read the methods at this point in the paper, so clarity here is especially important.

Our response: Thank you for your comment. You interpreted correctly the term modern time slice. It corresponds to the samples from sediments dated between 0 to 2000 cal yrs BP.

New text: line 174

Reviewer1 minor comment: Fig. 2 It seems weird to include carnivores (Homotherium, Panthera) in this figure. I would explain the justification or focus just on herbivores.

Our response: Thank you for your comment. The extinct mammals in question were predators of megaherbivores. We believe this underscores the occurrence of co-extinction events within the mammoth steppe ecosystem, closely linked with the extinction of megafauna. This aspect aligns well with the main theme of our study, which focuses on highlighting further instances of co-extinction, this time within the Viridiplantae kingdom.

Reviewer1 minor comment: 289 Possibly, though there are plenty of counter-examples of rare taxa that have survived as rare taxa for a very long time, and many abundant plant taxa went extinct in Europe (see: Svenning and Skov’s work on European tree extinctions).

Our response: Thank you for your comment. We now better nuance this and highlight that counter examples exist.

New text: line 472 to 474.

Reviewer1 minor comment: 293 The use of present tense here and in preceding sentences muddies the waters a bit. I would suggest past tense for results, which will make it more clear that you are talking about your results and not generalisms.

Our response: Thank you for your comment. We restructured and rewrote the entire section to improve the clarity.

New text: line 458 to 494

Reviewer1 minor comment: 303 This is potentially significant, and so I would like to see some discussion about why that is. Is it a rarefaction artifact (i.e., there were more forb and graminoids, so they are statistically likelier to go extinct)?

Our response: Thank you, this is a valuable comment. We changed the figure 4 and it now highlights the proportion of missing taxa per functional group relative to the total number of taxa. However, in supplementary figure 10, you can see the same figure but relative to the number of non-dbtaxa for each functional group. You can see that ~70% of non-dbtaxa that are grasses and ~55% that are forbs are likely to be absent from modern times. This is much higher than the non-dbtaxa shrubs and trees.

Reviewer1 minor comment: 315 This is a crucial point that should be highlighted sooner in the text, but it also needs some qualification, because the databases themselves are incomplete, which is a point that is glossed over in this text.

Our response: Thank you for your comment. We now introduce it better in the introduction and highlight the limitations and robustness of our approach in the first paragraph of the discussion to not leave ambiguities regarding the significance of our main results discussed.

New text: Line 95 to 98 and line 341 to 349.

Reviewer1 minor comment: 342 Or from Asparagaceae that just haven't been added yet.

Our response: Thank you for your comment. This possibility has been addressed and tested by examining taxa present in the Global Biodiversity Information Facility database (GBIF). This assessment aims to determine whether the lost non-dbtaxa represent only a fraction of taxa not covered by the reference database or if they signal potentially extinct taxa. To provide a comprehensive view, we now present both a minimum and maximum extinction rate, both of which exceed background rates. Additionally, in the first paragraph of the discussion section, we emphasize that the discovery of one of the taxa labelled as potentially extinct in fossil records or modern databases does not invalidate our conclusion regarding the percentage of extinct taxa. This is because new data may reveal additional taxa that could also be extinct.

New text: Figure1, line 182 to 196 and line 313 to 320.

357 This is another oversimplification that needs some unpacking or qualification. The underpinnings of ecological stability (or resistance and resilience) have been extensively studied (though questions still remain), and diverse ecosystems are often more stable even in the face of abiotic instability. And on longer timescales, environmental variability can be a speciation pump that drives diversification. We don't even know whether the tropics have more diversity because they are stable (museum) or because they drive diversification (cradle).

Our response: Thank you for your comment. As the text has been entirely restructured and rewritten. Still, we tried to be more nuanced as before the sentence was: "Abiotic stability is important for biotic stability" while we now say that our results provide an argument for this hypothesis, and not that it is universally known.

New text: line 442 to 444

Reviewer1 minor comment: 359 "It" has an unclear antecedent here.

Our response: This text has been entirely revised.

Reviewer1 minor comment: 362 Be careful about words like dramatic, which are normative. Also, how does this interpretation stand up to the fact that there have been multiple glacial cycles in the Quaternary? Shouldn't these be taxa that have already adapted to some degree to high amounts of environmental variability at high latitudes (both seasonally and on glacial-interglacial cycles)?

Our response: Thank you for your comment. We acknowledge that the Last Glacial to Holocene transition differs from previous glacial cycles of the Quaternary. While the mammoth steppe and megafauna managed to survive previous glacial cycles, they did not endure the transition to the Holocene. Further research is required to investigate past glacial and interglacial transitions and understand why megafauna and the mammoth steppe were able to withstand previous transitions but not the last one. Identifying similar mechanisms at play in these transitions could provide valuable insights.

Reviewer1 minor comment: 365-368 This is bordering on tautology – there was more turnover because there was more turnover.

Our response: Thank you for your comment. To avoid tautology, we reformulated.

New text: line 517 to 521

Reviewer1 minor comment: 369-371 This final sentence feels vague and underdeveloped – it's not entirely clear what the conclusion is meant to be.

Our response: Thank you. The final paragraph of the conclusion has been modified to clarify it.

New text: line 522 to 528

Reviewer1 minor comment: 375 Should be “Bering land bridge.” How do you differentiate between Beringia, Siberia, and Alaska (if Beringia is inclusive of both)?

Our response: Thank you. We refer to Beringia as covering Siberia and Alaska. We modified the text accordingly.

New text: line 532 to 533

Reviewer1 minor comment: 387 Some of these cores have been published previously – this statement implies that these are new cores. Is that the case, or was previously existing sediment used? If the latter, this needs qualifying.

Our response: Thank you for your comment. A site description is now provided in the supplementary note 1. Further, Table 1 points to references for the cores which previous work have been performed on (age models and *seDaDNA*).

Reviewer #2 (Remarks to the Author):

In their paper 'Potential plant extinctions with the loss of the Pleistocene mammoth-steppe' Courtin et al. use metabarcoding to investigate loss of plant species in arctic Russia and Alaska during the transition from the Pleistocene to the Holocene. They do so by characterizing ASVs according to their similarity to their reference database, and, in cases where ASVs appears distinct from both modern time slices and the reference database, they characterize that ASV as potentially extinct.

Reviewer2 major comment: I like the overall angle of the manuscript, and I agree with the authors that plant extinctions are terribly understudied. However, my main concern with this manuscript also relates to the focus on extinctions in the paper. I am not convinced that what the author quantify is actually extinctions. Could it not simply be range shifts, in combination with incomplete databases? I think the paper could benefit from a slight rewriting, with less emphasis on extinctions and more focus on community changes and perhaps loss of diversity.

Our response: Thank you for your comments. To improve the clarity of the study and underscore the robustness of our main finding regarding potential plant extinction, we have implemented additional tests to support our conclusion. We completely restructured and rewrote the text to ensure clarity. Moreover, we have standardized the vocabulary used to clearly convey that our study examines extinction rather than just extirpation. We conducted tests to verify that the fraction of lost non-dbtaxa did not solely result from missing taxa in the SibAla_2023 database. Additionally, we examined whether only a small fraction of the lost non-dbtaxa matched with other databases and, among the taxa matching, assessed the likelihood of their presence in refugia. In order to provide a more nuanced understanding, we now present both minimum and maximum extinction rates, both of which exceed background extinction rates. Still, as some limitations such as the limited number of sites and potential range shifts are present, we now highlight them in the first paragraph of the discussion, and discuss why our main conclusion, that plant extinction rate was low, is robust to those limitations.

New text: line 182 to 224 for the new tests to check robustness of our results. Line 305 to 385 for discussion of limitations.

Reviewer2 major comment: Furthermore, it might be beyond the scope of this paper, but the conclusions here could be strengthened by including more data. I know that the paper 'Fifty thousand years of Arctic vegetation and megafaunal diet' has a large trnL-gh dataset from the same area. Perhaps this dataset could be merged with the dataset at hand?

That being said, I think that this is an important paper with a novel approach to analyse these increasingly large and complicated metabarcoding datasets, and I think that it is a good fit for Nature communications.

Our response: Thank you for your comment. We appreciate your advice, and we agree that incorporating additional sites would enhance the robustness of our results. Consequently, we explored the possibility of merging our dataset with others from studies such as Wang et al.

2021, 10.1038/s41586-021-04016-x. However, none of the available datasets met the sample resolution required to apply our method effectively. For instance, the dataset provided by Wang et al. lacked samples that adequately covered our modern timeslice, and the sample resolution was insufficient to bin them into 2000-year intervals for investigating appearance and disappearance rates. We believe that the additional work prompted by the reviewers' comments significantly strengthens our conclusions.

Reviewer2 minor comment: Fig 1: I think that the map from figure 4 should be moved to this figure.

Our response: Thank you for your comment. As you may have noticed, the figures have been revised. Figure 1 now introduces the map and the methods applied, along with presenting the main results and tests conducted to highlight their robustness.

Reviewer2 minor comment: Furthermore, I would recommend splitting figure 1c into two panels, one for candidate taxa and one for dbtaxa. This would make it easier for the reader to appreciate subtle differences between the two. It is also currently a bit confusing that the same colours are repeated twice.

Our response: Thank you for your comment. This figure is now in the supplementary material as supplementary figure 3.

Reviewer2 minor comment: Line 140: This assumption is almost certainly incorrect. If you sampled more in the modern time slice, you might find many of these 'extirpated' taxa.

Our response: Thank you for your comment. We agree with this statement, although it applies to every taxon considered extinct. It is worth noting that some extinct taxa may reappear, despite our efforts to mitigate this possibility. We ensured uniform sampling across all time slices, and it is notable that most lost non-dbtaxa disappear from the dataset at 17,000- and 9,000-years BP. This makes it unlikely for these taxa to reappear. To address this, we developed an expected rate based on appearance and disappearance rates, considering that a taxon recently lost is more likely to reappear than one lost for multiple time slices. Furthermore, we conducted tests to verify that the fraction of missing non-dbtaxa from the modern time slice is not solely due to missing taxa from the reference database, by creating a synthetic dataset from taxa occurring in the Global Biodiversity Information Facility database (GBIF).

Reviewer2 minor comment: Line 146: 'the proportion of candidate taxa present in a time-slice decreases through time' I don't think that this is supported by the data in Fig 1c.

Our response: Thank you for your comment. This figure is now presented in the supplementary note 4.3. Further, in the samples from the Pleistocene (~27,000 to ~17,000 cal. yrs BP), 78.3% of the taxa were either forbs or graminoids, with more than 42% of the overall reads assigned. After the LGM, between ~15,000 and ~9,000 cal. yrs BP, there was a shift towards more shrub and tree taxa in the study area. Between ~7,000 and ~1,000 cal. yrs BP, 29.5% of all taxa are assigned to shrubs or trees with more than 91% of the overall reads assigned. For the dbtaxa we see a small decrease in forbs (73.5% of all dbtaxa before 15,000 cal. yrs BP and 71% after 9,000 cal. yrs BP) and graminoids (11.6% to 8.1%) between the

Pleistocene and the Holocene. The non-dbtaxa are generally more impacted with a decrease from 5.5% to 3.1% for graminoids and especially from 61.4% to 42.4% for forbs between before 15,000 cal. yrs BP and after 9,000 cal. yrs BP. This confirms that over time, non-dbtaxa are more often lost in comparison to dbtaxa.

Reviewer2 minor comment: Fig 2: It is not clear how the confidence threshold of 80% was chosen.

Our response: Thank you for your comment. We choose the 80th percentile of the distribution to show the variation but also to optimise visual aspects of the figure.

Reviewer2 minor comment: Furthermore, it would be interesting to also have the Greenland ice core data to compare with (perhaps overlaid on fig 2d).

Our response: Thank you for your comment. We tested several climate reconstruction methods, which are elaborated on in the supplementary note 6. These included simulated temperature and precipitation estimates from records covering the study area (Dallmeyer et al., 2022, 10.1038/s41467-022-33646-6), as well as temperature and precipitation estimates from pollen-based reconstructions (both MAT and WA-PLS, Herzshuh et al., 2022, 10.5194/essd-14-3213-2022). Our analysis indicated that both simulated and pollen-based reconstructions yielded similar results in terms of correlation to plant taxa loss. However, based on our experience and the fit of the data to expected changes in the region, we opted for pollen-based data over simulated data.

Reviewer2 minor comment: Fig 4: This figure gives a good overview of the analytical approach. Perhaps it should be moved, so that this becomes the new figure 1.

Our response: Thank you for your comment. This figure has been entirely reworked and is now placed as figure 1 following your recommendation.

Reviewer #3 (Remarks to the Author):

The authors have provided an important new contribution to our understanding of plant extinction and extirpation in the former “Mammoth Steppe” landscapes of the northern high latitudes. The geographic range of their study makes the work particularly valuable for scholars focusing on the Late Quaternary Extinctions and the patterns and processes driving that episode in Earth history. The dynamics of plant communities during and following the LQE are still not well understood. The authors have added new depth to this particular field by taking an expansive view across multiple Siberian/Alaskan biomes as they exist today. Their analysis of sedimentary DNA taken from lake cores drawn from multiple sites reveals significant patterns of plant losses correlating with known waves of large herbivore extinctions leading to the disappearance of the Mammoth Steppe. They find that rates of plant losses may have reached 120 times the background rates of extinction. Their results show these episodes correlate with rapid environmental shifts of the Late Pleistocene. Such instability becomes most pronounced in the high latitudes. From this, the authors make a good case for the relevance of their analysis of paleoecological proxy data, given that climate and environmental change in these regions is proceeding considerably more rapidly today. This report is building on a growing body of work in this area that is well-referenced here.

Reviewer3 minor comment: My comments below are minor but mainly focus on the clarity of the writing so as to make it more accessible to a wider body of readers. From the Abstract and in the Introduction and other sections, plant losses are characterized as “extirpations”, and “potential extinctions”, but these terms should be clearly defined and explained as they apply to the findings presented here. I am sure the authors have precise meanings for each of these terms, and I assume neither means actual extinction. It would be most helpful to the reader if these terms were explained early on in the Introduction. For the Abstract, I would suggest wording that briefly draws out the meaning of each, rather than introducing both terms parenthetically (line 19). Since the main thrust of the subsequent data interpretation hinges on these two concepts, it is important to have them made clear at the outset.

Our response: Thank you for your comment. We have made significant efforts to completely restructure and rewrite the introduction, results, and discussion sections of the main manuscript in order to reframe the study and enhance clarity for readers. Moreover, we believe that our study not only demonstrates extirpation but also potential extinction. We have conducted additional work to bolster the robustness of this main finding. Specifically, we have conducted new investigations to test whether the fraction of lost non-dbtaxa originated solely from missing taxa in the SibAla_2023 database. Furthermore, we have examined whether only a small fraction of the lost non-dbtaxa matched with other databases and, among the taxa matching, assessed their likelihood of presence in refugia. To add nuance to our study, we now provide both minimum and maximum extinction rates, both of which exceed background extinction rates. It's important to note that we maintain a conservative approach in detecting non-db and potentially extinct taxa.

New text: line 182 to 224 for the new analysis to test the robustness of our results and refine the potential extinction rates.

Reviewer3 minor comment: Beta diversity is another term that although used in certain areas of community ecology, is not commonly used in paleoecological studies. It would be helpful to the reader if the authors explained this concept early on in the Introduction, as it also is pivotal to their data interpretation.

Our response: Thank you for your comment. We now quickly introduce the concept in the introduction. Beta diversity is used to describe changes between community. It can be decomposed into several components to identify turnover between samples and contribution to communities.

New text: line 106 to 109

Reviewer3 minor comment: In the Discussion section, (line 262 and following) in referring to the two waves of megafaunal extinction, the first during the Late Glacial Maximum; and the second peaking between 15,000 and 10,000 years bp., we learn that “estimated plant extirpation” peaking around 13,000 bp, (in the middle of this time interval) does not correlate with megafaunal losses. Nevertheless, “potential plant extinction” does correlate with this second wave. At first, this seems paradoxical. No doubt this is an instance of the particular uses of the terms as outlined above in 1).

Our response: Thank you for your comment. In order to avoid any confusion, we have shifted our focus to emphasize the potential extinction signal which we consider the main finding of this study and have reduced the emphasis on the concept of extirpation. Additionally, we have highlighted the distinction between these two concepts in the introduction.

New text: line 85 to 87 for the introduction of both extinction and extirpation concepts.

Reviewer3 minor comment: It would be useful to have at least a brief description of the geographic setting of each of the sites where the cores were taken. As it is, the map in Figure 4 and Table 1 appear to be the only places where the reader can find this information, yet this would involve following each of the references listed in the far right column of Table 1.

Our response: Thank you for your comments. A description of the sites is now present in supplementary note 1 to complement Table 1.

Potential plant extinctions with the loss of the Pleistocene mammoth steppe

Response to the reviewers

Jérémy Courtin¹, Kathleen R. Stoof-Leichsenring¹, Simeon Lisovski¹, Ying Liu¹, Inger Greve Alsos², Boris K. Biskaborn¹, Bernhard Diekmann¹, Martin Melles⁵, Bernd Wagner⁵, Luidmila Pestryakova⁶, James Russell⁷, Yongsong Huang⁷ & Ulrike Herzschuh^{1,3,4*}

¹Polar Terrestrial Environmental Systems, Alfred Wegener Institute Helmholtz Centre for Polar and Marine Research, Germany

²The Arctic University Museum of Norway, UiT - The Arctic University of Norway, Norway

³Institute of Environmental Science and Geography, University of Potsdam, Germany

⁴Institute of Biology and Biochemistry, University of Potsdam, Germany

⁵Institute of Geology and Mineralogy, University of Cologne, Germany

⁶Institute of Natural Sciences, North-Eastern Federal University of Yakutsk, Russia

⁷Department of Earth, Environmental and Planetary Sciences, Brown University, USA

*Correspondence

Ulrike Herzschuh, ulrike.herzschuh@awi.de

Reviewer #2 (Remarks to the Author):

In their resubmitted article Courtin et al have used the GBIF database to estimate the expected number of absent species in their SibAla_2023 database. Using simulations based on this fraction the authors are able to estimate how large a proportion of the 'lost' taxa that are probably due to incomplete databases. Based on this, the authors estimate that 17 out of 60 species are potentially extinct. I like their approach of using simulations to account for the incomplete reference libraries, and, as such my main concern about the paper have been addressed.

Answer to reviewer: Thank you for your positive feedback and for endorsing our manuscript for publication. We appreciate your thoughtful comments, which helped us improve the clarity and overall quality of the paper. We are now glad the revisions have addressed your concerns, and we are grateful for your time and effort in reviewing our work.

I still have a couple of comments:

Maybe I have misunderstood how this simulation works, but the simulation suggests that 42/60 species are due to incomplete databases. What is the error margin on this estimate? Because if the error margin is, say +/- 20, there is no evidence for extinctions at all.

Answer to reviewer: Thank you for your comment. In the last revision round we included safeguards check to assess the likelihood of our results. This resulted in an estimation of the proportion of 60 taxa missing from the modern timeslice that are due to technicalities such as incomplete databases. You are correct that this resulted in an estimation that up to 42/60 taxa can be due to incomplete databases. 42 reflect the upper margin of our estimate.

Given that the main conclusion of the paper has been changed dramatically (i.e. from "extinction rate was up to 120X above bg" to "extinction rate were below modern estimates"), I think the title should be updated to reflect this. Perhaps the title could start with "Low extinction rates during the loss [...]" or "Plant resilience during the [...]"

Answer to reviewer: Thank you for your comment. We understand your perspective, however we believe that one of the key novelties of our study is demonstrating that, although plants were more resilient to extinction during the last glacial-interglacial transition compared to megafauna, our findings suggest that a fraction of plant taxa potentially have still undergone extinction. By highlighting this aspect in the title, we hope to spark further interest in the scientific community and encourage future research to explore this question in more depth. We also believe this study offers a novel method (among potentially many) to address the challenge of investigating past plant extinctions in the absence of macrofossils.

And speaking of the significant change in conclusion, I would like to flag that I am still not 100% convinced of the results. It is a very difficult task the authors have set themselves, estimating species loss. Having worked with metabarcoding data myself, I have sometimes found that seemingly small changes in the filtering stage (for example changing the cutoff from 100 read to 10 reads) can have dramatic effects on downstream analysis. But I guess only time will tell if these conclusions stand in the future.

Answer to reviewer: Thank you for your comment. We agree that only time will tell if the conclusions presented in this study stand in the future. As mentioned before, we hope that this study can lead to future research efforts that will further test our original findings and hypotheses.

Minor comments:

My last comment on what is now Supplementary fig 3 still stands. To me it does not make sense to show the non-db and db data stacked in this way, as it veils all patterns for the top category ('db taxa' in this case).

Kind regards,

Frederik Seersholm, University of Copenhagen

Answer to reviewer: Thank you for your comment. For us, this figure highlights that over time the fraction of non-db taxa decreases (bold line). However, we agree that the stack was not ideal to highlight pattern changes for the dbtaxa category and we changed the supplementary figure 3 following your recommendations.

Reviewer #2 (Remarks on code availability):

Code was not available for review as the github depositories are not publicly available (yet). It should be noted that according to the authors "they can be provided upon request" during the review process.

Answer to reviewer: Thank you for your comment. The scripts were not yet made publicly available on Github but were provided for the revision process as 2 "Related Manuscript File Supplementary Dataset" folders as Script pack 1 and Script pack 2. They are now made available to the public for the publication of the manuscript deposited under accession: <https://doi.org/10.5281/zenodo.14033298> and <https://doi.org/10.5281/zenodo.14033305>.

Reviewer #3 (Remarks to the Author):

The authors have addressed each one of my comments and concerns. They have reworked the text accordingly.

Reviewer 1's comments are incisive and thorough, highlighting some fundamental weaknesses in the structure of the author's overall argument and how the results have been brought to bear in support of that argument. Reviewer 1 nevertheless expresses enthusiasm for the ambitious aims and innovative approach to a complex problem. In

this, I agree; I find the revised manuscript better positioned to begin tackling that problem.

Regarding the major comments, I find the authors have addressed each one appropriately, restructuring their argument and engaging a broader body of the literature. Importantly they have incorporated the called for caveats and qualifications. The limitations of using existing reference databases and their way of addressing this are discussed in the paragraph following line 350.

The problem of the plant extinction window of 19,000 to 9,000 BP coinciding with the highest rate of climate-driven range shifts is addressed in 511ff where the authors identify a significant time lag before potential extinctions happen.

Noting the comment on the circularity of pollen-based temperature reconstructions to explain taxon losses derived from ancient DNA, I find the authors have made a good case for retaining this temperature reconstruction. While it is true that both proxies derive from plants, it is not the same array of plants. As observed by Jorgensen, et al. 2012, pollen, macrofossil, and sedaDNA data are complementary rather than overlapping.

Among the line-by-line remarks, I agree that mammoth steppe should not be hyphenated. Dale Guthrie coined the term and did not use a hyphen (although the title of Murchie, et al. 2021 uses a hyphen in Nature Communications).

I find the revised manuscript to be suitable for Nature Communications.

Answer to reviewer: Thank you very much for your thoughtful and constructive feedback. We greatly appreciate the time and effort you dedicated to reviewing our manuscript, as your insights have been invaluable in enhancing both the clarity and quality of our work. The comments provided have not only highlighted areas for improvement but have also guided us in refining the manuscript to better convey our findings. We are confident that these revisions have strengthened the overall contribution of our study, and we are sincerely grateful for your input throughout this process.